# Evaluating Gradient Inversion Attacks and Defenses in Federated Learning

**Yangsibo Huang**
Princeton University
Princeton, NJ 08540
yangsibo@princeton.edu

**Samyak Gupta**
Princeton University
Princeton, NJ 08540
samyakg@cs.princeton.edu

**Zhao Song**
Adobe Research
San Jose, CA 95110
zsong@adobe.com

**Kai Li**
Princeton University
Princeton, NJ 08540
li@cs.princeton.edu

**Sanjeev Arora**
Princeton University
Princeton, NJ 08540
arora@cs.princeton.edu

## Abstract

Gradient inversion attack (or input recovery from gradient) is an emerging threat to the security and privacy preservation of Federated learning, whereby malicious eavesdroppers or participants in the protocol can recover (partially) the clients' private data. This paper evaluates existing attacks and defenses. We find that some attacks make strong assumptions about the setup. Relaxing such assumptions can substantially weaken these attacks. We then evaluate the benefits of three proposed defense mechanisms against gradient inversion attacks. We show the trade-offs of privacy leakage and data utility of these defense methods, and find that combining them in an appropriate manner makes the attack less effective, *even under the original strong assumptions*. We also estimate the computation cost of end-to-end recovery of a single image under each evaluated defense. Our findings suggest that the state-of-the-art attacks can currently be defended against with minor data utility loss, as summarized in a list of potential strategies.

## 1 Introduction

Federated learning [McMahan et al., 2016, Kairouz et al., 2021] is a framework that allows multiple clients in a distributed environment to collaboratively train a neural network model at a central server, without moving their data to the central server. At every training step, each client computes a model update —i.e., gradient— on its local data using the latest copy of the global model, and then sends the gradient to the central server. The server aggregates these updates (typically by averaging) to construct a global model, and then sends the new model parameters to all clients. By allowing clients to participate in training without directly sharing their data, such protocols align better with data privacy regulations such as Health Insurance Portability and Accountability Act (HIPPA) [Act, 1996], California Consumer Privacy Act (CCPA) [Legislature, 2018], and General Data Protection Regulation [Commission, 2018].

While sharing gradients was thought to leak little information about the client's private data, recent papers [Zhu et al., 2019, Zhao et al., 2020, Geiping et al., 2020, Yin et al., 2021] developed a "gradient inversion attack" by which an attacker eavesdropping on a client's communications with the server can begin to reconstruct the client's private data. The attacker can also be a malicious participant in the Federated Learning scheme, including a honest-but-curious server who wishes to reconstruct private data of clients, or a honest-but-curious client who wishes to reconstruct private data of other clients. These attacks have been shown to work with batch sizes only up to 100 but even so they have

35th Conference on Neural Information Processing Systems (NeurIPS 2021).

created doubts about the level of privacy ensured in Federated Learning. The current paper seeks to evaluate the risks and suggest ways to minimize them.

Several defenses against gradient inversion attacks have been proposed. These include perturbing gradients [Zhu et al., 2019, Wei et al., 2020] and using transformation for training data that clients can apply on the fly [Zhang et al., 2018a, Huang et al., 2020]. More traditional cryptographic ideas including secure aggregation [Bonawitz et al., 2016] or homomorphic encryption [Phong et al., 2018] for the gradients can also be used and presumably stop any eavesdropping attacks completely. They will not be studied here due to their special setups and overhead.

We are not aware of a prior systematic evaluation of the level of risk arising from current attacks and the level of security provided by various defenses, as well as the trade-off (if any) between test accuracy, computation overhead, and privacy risks.

The paper makes two main contributions. First, we draw attention to two strong assumptions that a current gradient inversion attack [Geiping et al., 2020] implicitly makes. We show that by nullifying these assumptions, the performance of the attack drops significantly and can only work for low-resolution images. The findings are explored in Section 3 and already imply some more secure configurations in Federated Learning (Section 6).

Second, we summarize various defenses (Section 4) and systematically evaluate (Section 5) some of their performance of defending against a state-of-the-art gradient inversion attack, and present their data utility and privacy leakage trade-offs. We estimate the computation cost of end-to-end recovery of a single image under each evaluated defense. We also experimentally demonstrate the feasibility and effectiveness of combined defenses. Our findings are summarized as strategies to further improve Federated Learning's security against gradient inversion attacks (Section 6).

In Appendix B, we provide theoretical insights for mechanism of each evaluated defense.

## 2 Gradient Inversion Attacks

Previous studies have shown the feasibility of recovering input from gradient (i.e. gradient inversion) for image classification tasks, by formulating it as an optimization problem: given a neural network with parameters $\theta$, and the gradient $\nabla_\theta \mathcal{L}_\theta(x^*, y^*)$ computed with a private data batch $(x^*, y^*) \in \mathbb{R}^{b \times d} \times \mathbb{R}^b$ ($b, d$ being the batch size, image size), the attacker tries to recover $x \in \mathbb{R}^{b \times d}$, an approximation of $x^*$:

$$\arg\min_x \mathcal{L}_{\text{grad}}(x; \theta, \nabla_\theta \mathcal{L}_\theta(x^*, y^*)) + \alpha \mathcal{R}_{\text{aux}}(x) \tag{1}$$

The optimization goal consists of two parts: $\mathcal{L}_{\text{grad}}(x; \theta, \nabla_\theta \mathcal{L}_\theta(x^*, y^*))$ enforces matching of the gradient of recovered batch $x$ with the provided gradients $\mathcal{L}_\theta(x^*, y^*)$, and $\mathcal{R}_{\text{aux}}(x)$ regularizes the recovered image based on image prior(s).

[Phong et al., 2017] brings theoretical insights on this task by proving that such reconstruction is possible with a single-layer neural network. [Zhu et al., 2019] is the first to show that accurate pixel-level reconstruction is practical for a maximum batch size of 8. Their formulation uses $\ell_2$-distance as $\mathcal{L}_{\text{grad}}(\cdot, \cdot)$ but no regularization term $\mathcal{R}_{\text{aux}}(x)$. The approach works for low-resolution CIFAR datasets [Krizhevsky et al., 2009], with simple neural networks with sigmoid activations, but cannot scale up to high-resolution images, or larger models with ReLU activations. A follow-up [Zhao et al., 2020] proposes a simple approach to extract the ground-truth labels from the gradient, which improves the attack but still cannot overcome its limitations.

With a careful choice of $\mathcal{L}_{\text{grad}}$ and $\mathcal{R}_{\text{aux}}(x)$, [Geiping et al., 2020] substantially improves the attack and succeeds in recovering a single ImageNet [Deng et al., 2009] image from gradient: their approach uses cosine distance as $\mathcal{L}_{\text{grad}}$, and the total variation as $\mathcal{R}_{\text{aux}}(x)$. Their approach is able to reconstruct low-resolution images with a maximum batch size of 100 or a single high-resolution image. Based on [Geiping et al., 2020], [Wei et al., 2020] analyzes how different configurations in the training may affect attack effectiveness.

A more recent work [Yin et al., 2021] further improves the attack on high-resolution images, by introducing to $\mathcal{R}_{\text{aux}}(x)$ a new image prior term based on batch normalization [Ioffe and Szegedy, 2015] statistics, and a regularization term which enforces consistency across multiple attack trials.

An orthogonal line of work [Zhu and Blaschko, 2021] proposes to formulate the gradient inversion attack as a closed-form recursive procedure, instead of an optimization problem. However, their implementation can recover only low-resolution images under the setting where batch size = 1.

## 3 Strong Assumptions Made by SOTA Attacks

### 3.1 The state-of-the-art attacks

Two recent attacks [Geiping et al., 2020, Yin et al., 2021] achieve best recovery results. Our analysis focuses on the former as the implementation of the latter is not available at the time of writing this paper. We plan to include the analysis for the latter attack in the final version of this paper if its implementation becomes available.

[Geiping et al., 2020]'s attack optimizes the following objective function:

$$\arg\min_x 1 - \frac{\langle \nabla_\theta \mathcal{L}_\theta(x, y), \nabla_\theta \mathcal{L}_\theta(x^*, y^*) \rangle}{\| \nabla_\theta \mathcal{L}_\theta(x, y) \| \| \nabla_\theta \mathcal{L}_\theta(x^*, y^*) \|} + \alpha_{\mathrm{TV}} \mathcal{R}_{\mathrm{TV}}(x) \tag{2}$$

where $\langle \cdot, \cdot \rangle$ is the inner-product between vectors, and $\mathcal{R}_{\mathrm{TV}}(\cdot)$ is the total variation of images.

We notice that Geiping et al. has made two strong assumptions (Section 3.2). Changing setups to invalidate those assumptions will substantially weaken the attacks (Section 3.3). We also summarize whether other attacks have made similar assumptions in Table 1.

### 3.2 Strong assumptions

We find that previous gradient inversion attacks have made different assumptions about whether the attacker knows Batch normalization statistics or private labels, as shown in Table 3.2. Note that [Geiping et al., 2020]'s attack makes both strong assumptions.

**Assumption 1: Knowing BatchNorm statistics.** Batch normalization (BatchNorm) [Ioffe and Szegedy, 2015] is a technique for training neural networks that normalizes the inputs to a layer for every mini-batch. It behaves differently during training and evaluation. Assume the model has $L$ batch normalization layers. Given $x^*$, a batch of input images, we use $x_l^*$ to denote the input features to the $l$-th BatchNorm layer, where $l \in [L]$. During training, the $l$-th BatchNorm layer normalizes $x_l^*$ based on the batch's mean $\mathrm{mean}(x_l^*)$ and variance $\mathrm{var}(x_l^*)$, and keeps a running estimate of mean and variance of all training data points, denoted by $\mu_l$ and $\sigma_l^2$. During inference, $\{\mu_l\}_{l=1}^L$ and $\{\sigma_l^2\}_{l=1}^L$ are used to normalize test images. In the following descriptions, we leave out $\{\cdot\}_{l=1}^L$ for simplicity (i.e. use $\mu, \sigma^2$ to denote $\{\mu_l\}_{l=1}^L, \{\sigma_l^2\}_{l=1}^L$, and $\mathrm{mean}(x^*), \mathrm{var}(x^*)$ to denote $\{\mathrm{mean}(x_l^*)\}_{l=1}^L, \{\mathrm{var}(x_l^*)\}_{l=1}^L$).

We notice that [Geiping et al., 2020]'s implementation[1] assumes that BatchNorm statistics of the private batch, i.e., $\mathrm{mean}(x^*), \mathrm{var}(x^*)$, are jointly provided with the gradient. Knowing BatchNorm statistics would enable the attacker to apply the same batch normalization used by the private batch on his recovered batch, to achieve a better reconstruction. This implicitly increases the power of the attacker, as sharing private BatchNorm statistics are not necessary in Federated learning [Andreux et al., 2020, Li et al., 2021].

Note that this assumption may be *realistic* in some settings: 1) the neural network is shallow, thus does not require using BatchNorm layers, or 2) the neural network is deep, but adapts approaches that normalize batch inputs with a fixed mean and variance (as alternative to BatchNorm), e.g. Fixup initialization [Zhang et al., 2019].

**Assumption 2: Knowing or able to infer private labels.** Private labels are not intended to be shared in Federated learning, but knowing them would improve the attack. [Zhao et al., 2020] finds that label information of a *single* private image can be inferred from the gradient (see Section 3.3 for details). Based on this, [Geiping et al., 2020] assumes the attacker knows private labels (see remark at the end of Section 4 in their paper). However, this assumption may not hold true when multiple images in a batch share the same label, as we will show in the next section.

---

[1]The official implementation of [Geiping et al., 2020]: https://github.com/JonasGeiping/invertinggradients.

| Assumptions | [Zhu et al., 2019] | [Zhao et al., 2020] | [Geiping et al., 2020] | [Yin et al., 2021] |
|---|---|---|---|---|
| **Knowing BN statistics** | N/A[†] | N/A[†] | Yes | Yes[*] |
| **Knowing private labels** | No | No[‡] | Yes | No[‡] |

Table 1: Assumptions of gradient inversion attacks. [†]: its evaluation uses a simple model without a BatchNorm layer; [‡]: it proposes a method to infer private labels, which works when images in a batch have unique labels (see Section 3.3); [*]: although the paper discusses a setting where BatchNorm statistics are unknown, its main results assume knowing BatchNorm statistics.

### 3.3 Re-evaluation under relaxed assumptions

We re-evaluate the performance of the gradient inversion attack in settings where two assumptions above are relaxed. For each relaxation, we re-design the attack (if needed) based on the knowledge that the attacker has.

**Relaxation 1: Not knowing BatchNorm statistics.** We refer to the previous threat model as $\mathrm{BN_{exact}}$, where the attacker knows exact BatchNorm statistics of the private batch. We consider a more realistic threat model where these statistics are not exposed, and re-design the attack based on it.

*Threat model.* In each training step, the client normalizes its private batch $x^*$ using the batch's mean $\mathrm{mean}(x^*)$ and variance $\mathrm{var}(x^*)$, keeps the running estimate of mean and variance *locally* as in [Li et al., 2021], and shares the gradient. The client releases the final aggregated mean $\mu$, and aggregated variance $\sigma^2$ of all training data points at the end of training. Same as before, the attacker has access to the model and the gradient during training.

*Re-design A:* $\mathrm{BN_{proxy}}$, *attacker naively uses $\mu$ and $\sigma^2$.* A simple idea is that the attacker uses $(\mu, \sigma^2)$ as the proxy for $(\mathrm{mean}(x^*), \mathrm{var}(x^*))$, and uses them to normalize $x$, his guesses of the private batch. Other operations of the gradient inversion attack remain the same as before. However, Figure 1.d and 1.h show poor-quality reconstruction with this re-design.

*Re-design B:* $\mathrm{BN_{infer}}$, *attacker infers $(\mathrm{mean}(x^*), \mathrm{var}(x^*))$ based on $(\mu, \sigma^2)$.* A more reasonable attacker will try to infer $(\mathrm{mean}(x^*), \mathrm{var}(x^*))$ while updating $x$, his guesses of the private batch, and uses $(\mathrm{mean}(x), \mathrm{var}(x))$ to normalize the batch. In this case, $(\mu, \sigma^2)$ could be used as a prior of BatchNorm statistics to regularize the recovery, as suggested in [Yin et al., 2021]:

$$\arg \min_x 1 - \frac{\langle \nabla_\theta \mathcal{L}_\theta(x,y), \nabla_\theta \mathcal{L}_\theta(x^*,y^*) \rangle}{\|\nabla_\theta \mathcal{L}_\theta(x,y)\| \| \nabla_\theta \mathcal{L}_\theta(x^*,y^*)\|} + \alpha_{\mathrm{TV}} \mathcal{R}_{\mathrm{TV}}(x) + \alpha_{\mathrm{BN}} \mathcal{R}_{\mathrm{BN}}(x) \qquad (3)$$

where $\mathcal{R}_{\mathrm{BN}}(x) = \sum_l \|\mathrm{mean}(x_l) - \mu_l\|_2 + \sum_l \|\mathrm{var}(x_l) - \sigma_l^2\|_2$.

We tune $\alpha_{\mathrm{BN}}$ and present the best result in Figure 1.c and 1.g (see results of different $\alpha_{\mathrm{BN}}$'s in Appendix A). As shown, for a batch of low-resolution images, $\mathrm{BN_{infer}}$ gives a much better reconstruction result than $\mathrm{BN_{proxy}}$, but still cannot recover some details of the private batch when compared with $\mathrm{BN_{exact}}$. The result for a single high-resolution image is worse: the attacker fails to return a recognizable reconstruction with $\mathrm{BN_{infer}}$. This suggests not having access to BatchNorm statistics of the private batch already weakens the state-of-the-art gradient inversion attack.

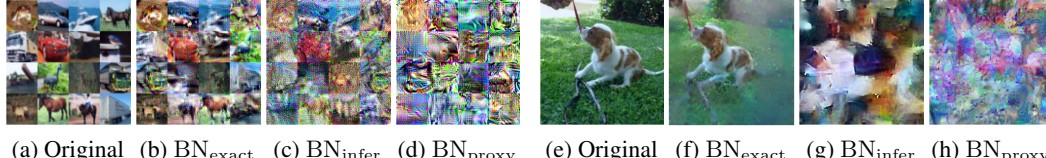

(a) Original  (b) $\mathrm{BN_{exact}}$  (c) $\mathrm{BN_{infer}}$  (d) $\mathrm{BN_{proxy}}$  (e) Original  (f) $\mathrm{BN_{exact}}$  (g) $\mathrm{BN_{infer}}$  (h) $\mathrm{BN_{proxy}}$

Figure 1: Attacking a batch of 16 low-resolution images from CIFAR-10 (a-d) and a single high-resolution image from ImageNet (e-h) with different knowledge of BatchNorm statistics. Attack is weakened when BatchNorm statistics are not available (c, d versus b, and g, h versus f). See Appendix A for more examples and quantitative results.

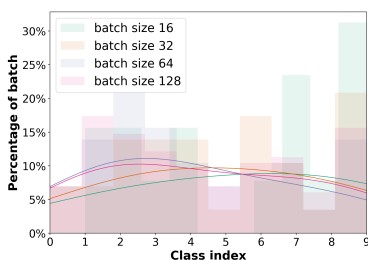
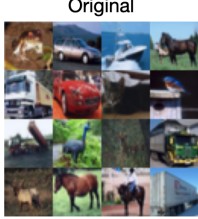
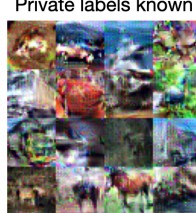
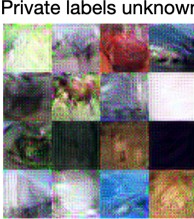

Original     Private labels known     Private labels unknown

(a) Distribution of labels in a batch        (b) Reconstructions with and without private labels

Figure 2: Attack is weakened when private labels are not available. (a) shows that for CIFAR-10, when the batch size is large, many images in the batch belong to the same class, which essentially weakens label restoration [Zhao et al., 2020, Yin et al., 2021]. (b) visualizes a reconstructed batch of 16 images with and without private labels known. The quality of the reconstruction drops without knowledge of private labels.

**Relaxation 2: Not knowing private labels.**  [Zhao et al., 2020] notes that label information of a *single* private image can be computed analytically from the gradients of the layer immediately before the output layer. [Yin et al., 2021] further extends this method to support recovery of labels for a batch of images. However, if multiple images in the private batch belong to a same label, neither approach can tell how many images belong to that label, let alone which subset of images belong to that label. Figure 2a demonstrates that with CIFAR-10, for batches of various sizes it is possible for many of the training samples to be have the same label, and the distribution of labels is not uniform - and hence, inferring labels becomes harder and the attack would be weakened. In Figure 2b we evaluate the worst-case for an attacker in this setting by comparing recoveries where the batch labels are simultaneously reconstructed alongside the training samples.

# 4  Defenses Against the Gradient Inversion Attack

Several defense ideas have been proposed to mitigate the risks of gradient inversion.

## 4.1  Encrypt gradients

Cryptography-based approaches encrypt gradient to prevent gradient inversion. [Bonawitz et al., 2016] presents a secure aggregation protocol for Federated learning by computing sum of gradient vectors based on secret sharing [Shamir, 1979]. [Phong et al., 2018] proposes using homomorphic encryption to encrypt the gradients before sending. These approaches require special setup and can be costly to implement.

Moreover, with secure aggregation protocol, an honest-but-curious server can still launch the gradient inversion attack on the summed gradient vector. Similarly, an honest-but-curious client can launch the gradient inversion attack on the model returned by the server to reconstruct other clients' private data, even with homomorphic encryption.

As alternatives, two other types of defensive mechanisms have been proposed to mitigate the risks of attacks on *plain-text* gradient.

## 4.2  Perturbing gradients

**Gradient pruning.**  When proposing the first practical gradient inversion attack, [Zhu et al., 2019] also suggests a defense by setting gradients of small magnitudes to zero (i.e. gradient pruning). Based on their attack, they demonstrate that pruning more than 70% of the gradients would make the recovered images no longer visually recognizable. However, the suggested prune ratio is determined based on weaker attacks, and may not remain safe against the state-of-the-art attack.

**Adding noise to gradient.**  Motivated by DPSGD [Abadi et al., 2016] which adds noise to gradients to achieve differential privacy [Dwork, 2009, Dwork and Roth, 2014], [Zhu et al., 2019, Wei et al.,

2020] also suggests defending by adding Gaussian or Laplacian noise to gradient. They show that a successful defense requires adding high noise level such that its accuracy drops by more than $30\%$ with CIFAR-10 tasks. Recent works [Papernot et al., 2020, Tramèr and Boneh, 2021] suggest using better pre-training techniques and a large batch size (e.g. $4,096$) to achieve a better accuracy for DPSGD training.

Since most DPSGD implementations for natural image classification tasks [Abadi et al., 2016, Papernot et al., 2020, Tramèr and Boneh, 2021] use a pre-training and fine-tuning pipeline, it is hard to fairly compare with other defense methods that can directly apply when training the model from scratch. Thus, we leave the comparison with DPSGD to future work.

### 4.3 Weak encryption of inputs (i.e. encoding inputs)

**MixUp.** MixUp data augmentation [Zhang et al., 2018a] trains neural networks on composite images created via linear combination of image pairs. It has been shown to improve the generalization of the neural network and stabilizes the training. Recent work also suggests that MixUp increases the model's robustness to adversarial examples [Pang et al., 2020, Lamb et al., 2019].

**InstaHide.** Inspired by MixUp, [Huang et al., 2020] proposes InstaHide as a light-weight instance-encoding scheme for private distributed learning. To encode an image $x \in \mathbb{R}^d$ from a private dataset, InstaHide first picks $k-1$ other images $s_2, s_3, \ldots, s_k$ from that private dataset, or a large public dataset, and $k$ random nonnegative coefficients $\{\lambda_i\}_{i=1}^k$ that sum to 1, and creates a composite image $\lambda_1 x + \sum_{i=2}^k \lambda_i s_i$ ($k$ is typically small, e.g., $4$). A composite label is also created using the same set of coefficients.[2] Then it adds another layer of security: pick a random sign-flipping pattern $\sigma \in \{-1, 1\}^d$ and output the encryption $\tilde{x} = \sigma \circ (\lambda_1 x + \sum_{i=2}^k \lambda_i s_i)$, where $\circ$ is coordinate-wise multiplication of vectors. The neural network is then trained on encoded images, which look like random pixel vectors to the human eye and yet lead to good classification accuracy ($< 6\%$ accuracy loss on CIFAR-10, CIFAR-100, and ImageNet).

Recently, [Carlini et al., 2020] gives an attack to recover private images of a small dataset, when the InstaHide encodings are revealed to the attacker (not in a Federated learning setting). Their first step is to train a neural network on a public dataset for similarity annotation, to infer whether a pair of InstaHide encodings contain the same private image. With the inferred similarities of all pairs of encodings, the attacker then runs a combinatorial algorithm (cubic time in size of private dataset) to cluster all encodings based on their original private images, and finally uses a regression algorithm (with the help of composite labels) to recover the private images.

Neither [Huang et al., 2020] or [Carlini et al., 2020] has evaluated their defense or attack in the Federated learning setting, where the attacker observes gradients of the encoded images instead of original encoded images. This necessitates the systematic evaluation in our next section.

## 5 Evaluation of defenses

The main goal of our experiments is to understand the trade-offs between data utility (accuracy) and securely defending the state-of-the-art gradient inversion attack even in its strongest setting, *without* any relaxation of its implicit assumptions. Specifically, we grant the attacker the knowledge of 1) BatchNorm statistics of the private batch, and 2) labels of the private batch.

We vary key parameters for each defense, and evaluate their performance in terms of the test accuracy, computation overhead, and privacy risks (Section 5.2). We then investigate the feasibility of combining defenses (Section 5.3). We also estimate the computation cost of end-to-end recovery of a single image under evaluated defenses (Section 5.4).

As the feasibility of the state-of-the-art attack [Geiping et al., 2020] on a batch of high-resolution images remain elusive when its implicit assumptions no longer hold (see Figure 1), we focus on the evaluation with low-resolution in trying to understand whether current attacks can be mitigated.

---

[2]Only the labels of examples from the private dataset will get combined. See [Huang et al., 2020] for details.

## 5.1 Experimental setup

**Key parameters of defenses.** We evaluate following defenses on CIFAR-10 dataset [Krizhevsky et al., 2009] with ResNet-18 architecture [He et al., 2016].

- **GradPrune** (gradient pruning): gradient pruning set gradients of small magnitudes to zero. We vary the pruning ratio $p$ in $\{0.5, 0.7, 0.9, 0.95, 0.99, 0.999\}$.

- **MixUp**: MixUp encodes a private image by linearly combining it with $k-1$ other images from the training set. Following [Huang et al., 2020], we vary $k$ in $\{4, 6\}$, and set the upper bound of a single coefficient to $0.65$ (coefficients sum to 1).

- **Intra-InstaHide**: InstaHide [Huang et al., 2020] proposes two versions: Inter-InstaHide and Intra-InstaHide. The only difference is that at the mixup step, Inter-Instahide mixes up an image with images from a public dataset, whereas Intra-InstaHide only mixes with private images. Both versions apply a random sign flipping pattern on each mixed image. We evaluate Intra-InstaHide in our experiments, which is a weaker version of InstaHide. Similar to the evaluation of MixUp, we vary $k$ in $\{4, 6\}$, and set the upper bound of a single coefficient to $0.65$. Note that InstaHide flips signs of pixels in the image, which destroys the total variation prior. However, the absolute value of adjacent pixels should still be close. Therefore, for the InstaHide defense, we apply the total variation regularizer on $|x|$, i.e. taking absolute value of each pixel in the reconstruction.

We train the ResNet-18 architecture on CIFAR-10 using different defenses, and launch the attack. We provide more details of the experiments in Appendix A.

**The attack.** We use a subset of 50 CIFAR-10 images to evaluate the attack performance. Note that attacking MixUp and InstaHide involves another step to decode private images from the encoded images. We apply [Carlini et al., 2020]'s attack here as the decode step, where the attacker needs to eavesdrop $T$ epochs of training, instead of a single training step. We set $T = 20$ in our evaluation. We also grant the attacker the strongest power for the decode step to evaluate the upper bound of privacy leakage. Given a MixUp or Intra-InstaHide image which encodes $k$ private images, we assume the attacker knows:

1. The indices of $k$ images in the private dataset. In a realistic scenario, the attacker of [Carlini et al., 2020] would need to train a neural network to detect similarity of encodings, and run a combinatorial algorithm to solve *an approximation of* this mapping.

2. The mixing coefficients for each of the $k$ private image. In real Federated learning, this information is *not available*.

**Hyper-parameters of the attack.** The attack minimize the objective function given in Eq.3. We search $\alpha_{\mathrm{TV}}$ in $\{0, 0.001, 0.005, 0.01, 0.05, 0.1, 0.5\}$ for all defenses, and apply the best choice for each defense: $0.05$ for GradPrune, $0.1$ for MixUp, and $0.01$ for Intra-InstaHide. We apply $\alpha_{\mathrm{BN}} = 0.001$ for all defenses after searching it in $\{0, 0.0005, 0.001, 0.01, 0.05, 0.01\}$. We optimize the attack for $10,000$ iterations using Adam [Kingma and Ba, 2015], with initial learning rate $0.1$. We decay the learning rate by a factor of $0.1$ at $3/8, 5/8, 7/8$ of the optimization.

**Batch size of the attack.** [Zhu et al., 2019, Geiping et al., 2020] have shown that a small batch size is important for the success of the attack. We intentionally evaluate the attack with three small batch sizes to test the upper bound of privacy leakage, including the minimum (and unrealistic) batch size 1, and two small but realistic batch sizes, 16 and 32.

**Metrics for reconstruction quality.** We visualize reconstructions obtained under different defenses. Following [Yin et al., 2021], we also use the learned perceptual image patch similarity (LPIPS) score [Zhang et al., 2018b] to measure mismatch between reconstruction and original images: higher values suggest more mismatch (less privacy leakage).

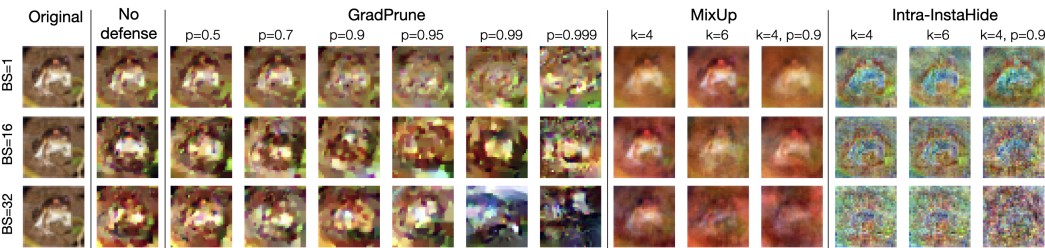

Figure 3: Reconstruction results under different defenses with batch size being 1, 16 and 32. When batch size is 32, combining gradient pruning and Intra-InstaHide makes the reconstruction almost unrecognizable (the last column). See Figure 7 in Appendix A for the full version.

| | None | GradPrune (p) | | | | | | MixUp (k) | | Intra-InstaHide (k) | | GradPrune (p = 0.9) | |
| | | | | | | | | | | | | + MixUp | + Intra-InstaHide |
|---|---|---|---|---|---|---|---|---|---|---|---|---|---|
| **Parameter** | - | 0.5 | 0.7 | 0.9 | 0.95 | 0.99 | 0.999 | 4 | 6 | 4 | 6 | $k = 4$ | $k = 4$ |
| **Test Acc.** | 93.37 | 93.19 | 93.01 | 90.57 | 89.92 | 88.61 | 83.58 | 92.31 | 90.41 | 90.04 | 88.20 | 91.37 | 86.10 |
| **Time (train)** | $1\times$ | $1.04\times$ | | | | | | $1.06\times$ | | $1.06\times$ | | $1.10\times$ | |
| **Attack batch size $= 1$** | | | | | | | | | | | | | |
| **Avg. LPIPS** ↓ | 0.19 | 0.19 | 0.22 | 0.35 | 0.42 | 0.52 | 0.52 | 0.34 | 0.46 | 0.58 | **0.61** | 0.41 | 0.60 |
| **Best LPIPS** ↓ | 0.02 | 0.02 | 0.05 | 0.14 | 0.22 | 0.32 | 0.36 | 0.12 | 0.25 | 0.41 | 0.42 | 0.21 | **0.43** |
| (LPIPS std.) | 0.16 | 0.17 | 0.16 | 0.13 | 0.11 | 0.08 | 0.06 | 0.08 | 0.07 | 0.06 | 0.09 | 0.07 | 0.09 |
| **Attack batch size $= 16$** | | | | | | | | | | | | | |
| **Avg. LPIPS** ↓ | 0.45 | 0.46 | 0.47 | 0.51 | 0.55 | 0.58 | 0.61 | 0.34 | 0.31 | 0.62 | 0.63 | 0.46 | **0.68** |
| **Best LPIPS** ↓ | 0.18 | 0.19 | 0.19 | 0.31 | 0.43 | 0.47 | 0.51 | 0.11 | 0.13 | 0.41 | 0.44 | 0.22 | **0.54** |
| (LPIPS std.) | 0.12 | 0.12 | 0.11 | 0.07 | 0.05 | 0.04 | 0.03 | 0.09 | 0.09 | 0.08 | 0.08 | 0.10 | 0.07 |
| **Attack batch size $= 32$** | | | | | | | | | | | | | |
| **Avg. LPIPS** ↓ | 0.45 | 0.46 | 0.48 | 0.52 | 0.54 | 0.58 | 0.63 | 0.50 | 0.49 | 0.69 | 0.69 | 0.62 | **0.73** |
| **Best LPIPS** ↓ | 0.18 | 0.18 | 0.22 | 0.31 | 0.43 | 0.48 | 0.54 | 0.31 | 0.28 | 0.56 | 0.56 | 0.37 | **0.65** |
| (LPIPS std.) | 0.11 | 0.11 | 0.09 | 0.07 | 0.05 | 0.04 | 0.04 | 0.10 | 0.10 | 0.06 | 0.07 | 0.10 | 0.05 |

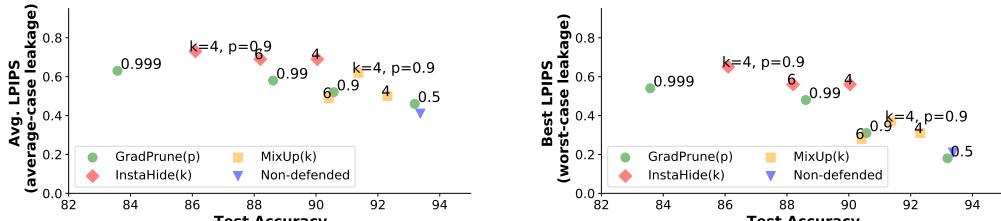

Table 2: Utility-security trade-off of different defenses. We train the ResNet-18 model on the whole CIFAR-10 dataset, and report the averaged test accuracy and running time of 5 independent runs. We evaluate the attack on a subset of 50 CIFAR-10 images, and report the LPIPS score (↓: lower values suggest more privacy leakage). We mark the least-leakage defense measured by the metric in **green**.

## 5.2 Performance of defense methods

We summarize the performance of each defense in Table 2, and visualize reconstructed images in Figure 3. We report the averaged and the best results for the metric of reconstruction quality, as a proxy for average-case and worst-case privacy leakage.

**No defense.** Without any defense, when batch size is 1, the attack can recover images well from the gradient. Increasing the batch size makes it difficult to recover well, but the recovered images are visually similar to the originals (see Figure 3).

**Gradient pruning (GradPrune).** Figure 3 shows that as the pruning ratio $p$ increases, there are more artifacts in the reconstructions. However, the reconstructions are still recognizable even when the pruning ratio $p = 0.9$, thus the previous suggestion of using $p = 0.7$ by [Zhu et al., 2019] is no longer safe against the state-of-the-art attack. Our results suggest that, for CIFAR-10, defending the

strongest attack with gradient pruning may require the pruning ratio $p \geq 0.999$. As a trade-off, such a high pruning ratio would introduce an accuracy loss of around $10\%$ (see Table 2).

**MixUp.** MixUp introduces a small computational overhead to training. MixUp with $k = 4$ only has a minor impact (~2%) on test accuracy, but it is not sufficient to defend the gradient inversion attack (see Figure 3). Increasing $k$ from 4 to 6 slightly reduces the leakage, however, the reconstruction is still highly recognizable. This suggests that MixUp alone may not be a practical defense against the state-of-the-art gradient inversion attack.

**Intra-InstaHide.** Intra-InstaHide with $k = 4$ incurs an extra ~2% accuracy loss compared with MixUp, but it achieves better defense performance: when batch size is 32, there are obvious artifacts and color shift in the reconstruction (see Figure 3). However, with batch size 32, Intra-InstaHide alone also cannot defend the state-of-the-art gradient inversion, as structures of private images are still vaguely identifiable in reconstructions.

Appendix A provides the whole reconstructed dataset under MixUp and Intra-InstaHide.

### 5.3 Performance of combined defenses

We notice that two types of defenses (i.e perturbing gradient and encoding inputs) are complementary to each other, which motivates an evaluation of combining gradient pruning with MixUp or Intra-InstaHide.

As shown in Figure 3, when the batch size is 32, combining Intra-InstaHide ($k = 4$) with gradient pruning ($p = 0.9$) makes the reconstruction almost unrecognizable. The combined defense yields a higher LPIPS score than using gradient pruning with $p = 0.999$, but introduces a smaller accuracy loss (~7% compared with a no-defense pipeline).

Note that our evaluation uses *the strongest attack* and *relatively small batch sizes*. As shown in Appendix A, invalidating assumptions in Section 3 or increasing the batch size may hinder the attack with an even weaker defense (e.g. with a lower $p$, or smaller $k$), which gives a better accuracy.

### 5.4 Time estimate for end-to-end recovery of a single image

Table 3 shows time estimates for the end-to-end recovery of a single image in a Federated learning setting with GradPrune or InstaHide defense. We do not estimate for MixUp since it has been shown to be a weak defense (see Section 5.2).

Our time estimates consider three fairly small dataset sizes. The largest size in our estimate is a small fraction of a dataset of ImageNet scale. We consider a client holds a dataset of $N$ private images and participates in Federated learning, which trains a ResNet-18 model with batch size $b = 128$. Assumes that the resolution of the client's data is $32 \times 32 \times 3$. If the attacker uses a single NVIDIA GeForce RTX 2080 Ti GPU as his computation resource, and runs gradient inversion with 10,000 iterations of optimization, then $t$, the running time for attacking a single batch is ~0.25 GPU hours (batch size $b$ has little impact on the attack's running time, but a larger $b$ makes the attack less effective).

**Non-defended and gradient pruning.** Recovering a single image in a non-defended pipeline (or a pipeline that applies gradient pruning alone as the defense) only requires the attacker to invert gradient of a single step of training, which takes time $t$.

**InstaHide.** When InstaHide is applied, the current attack [Carlini et al., 2020] suggests that recovering a single image would involve recovering the whole dataset first. As discussed in Section 4, Carlini et al.'s attack consists of two steps: 1) recover InstaHide images from gradient of $T$ epochs. This would take $(NT/b) \times t$ GPU hours. 2) Run the decode attack [Carlini et al., 2020] on InstaHide images to recover the private dataset, which involves:

2a Train a neural network to detect similarity in recovered InstaHide images. Assume that training the network requires at least $n$ recovered InstaHide images, then collecting these images by running gradient inversion would take $(n/b) \times t$ GPU hours. The training takes 10 GPU hours according to [Carlini et al., 2020], so training the similarity network would take $(n/b) \times t + 10$ GPU hours in total.

| Size of client's dataset ($N$) | No defense | GradPrune | InstaHide |
|:---:|:---:|:---:|:---:|
| 5,000 | | | 934.48 |
| 50,000 | 0.25 | 0.25 | 46,579.01 ($\approx$ 5.5 GPU years) |
| 500,000 | | | 4,215,524.32 ($\approx$ 493.4 GPU years) |

Table 3: Time estimates (NVIDIA GeForce RTX 2080 Ti GPU hours) of recovering *a single image* from the client's dataset using the state-of-the-art gradient inversion attack [Geiping et al., 2020] under different defenses. We assume image resolution of the client's data is $32 \times 32 \times 3$.

2b Run the combinotorial algorithm to recover original images. Running time of this step has been shown to be at least quadratic in $m$, the number of InstaHide encodings [Chen et al., 2021]. This step takes $1/6$ GPU hours with $m = 5 \times 10^3$. Therefore for $m = NT$, the running time is at least $1/6 \times (\frac{NT}{5 \times 10^3})^2$ GPU hours.

In total, an attack on InstaHide in this real-world setting would take $(NT/b) \times t + (n/b) \times t + 10 + 1/6 \times (\frac{NT}{5 \times 10^3})^2$ GPU hours. We use $T = 50$ (used by [Carlini et al., 2020]), $n = 10,000$ and give estimate in Table 3. As shown, when InstaHide is applied on a small dataset ($N = 5,000$), the end-to-end recovery of a single image takes $> 3,000\times$ longer than in a no-defense pipeline or GradPrune pipeline; when InstaHide is applied on a larger dataset ($N = 500,000$), the computation cost for end-to-end recovery is enormous.

## 6 Conclusions

This paper first points out that some state-of-the-art gradient inversion attacks have made strong assumptions about knowing BatchNorm statistics and private labels. Relaxing such assumptions can significantly weaken these attacks.

The paper then reports the performance of a set of proposed defenses against gradient inversion attacks, and estimates the computation cost of an end-to-end recovery of a single image in different dataset sizes. Our evaluation shows that InstaHide without mixing with data from a public dataset combined with gradient pruning can defend the state-of-the-art attack, and the estimated time to recover a single image in a medium-size client dataset (e.g. of 500,000 images) is enormous.

Based on our evaluation of the attack by [Geiping et al., 2020] and multiple defenses for *plain-text* gradients, we have the following observations:

- *Using BatchNorm layers in your deep net but don't share BatchNorm statistics of the private batch during Federated learning weakens the attack*. We have demonstrated in Section 3 that exposing BatchNorm statistics to the attacker significantly improves the quality of gradient inversion. So a more secure configuration of Federated Learning would be to use BatchNorm layers, but do not share BatchNorm statistics in training, which has been shown feasible in [Andreux et al., 2020, Li et al., 2021].

- *Using a large batch size weakens the attack; a batch size smaller than 32 is not safe*. We have shown that a larger batch size hinders the attack by making it harder to guess the private labels (Section 3) and to recover the private images even with correct private labels (Section 5). Our experiments suggest that even with some weak defenses applied, a batch size smaller than 32 is not safe against the strongest gradient inversion attack.

- *Combining multiple defenses may achieve a better utility-privacy trade-off*. In our experiment, for a batch size of 32, combining InstaHide ($k = 4$) with gradient pruning ($p = 0.9$) achieves the best utility-privacy trade-off, by making the reconstruction almost unrecognizable at a cost of ~7% accuracy loss (using InstaHide also makes the end-to-end recovery of a single image more computationally expensive). Best parameters would vary for different deep learning tasks, but we strongly encourage Federated learning participants to explore the possibility of combining multiple defensive mechanisms, instead of only using one of them.

We hope to extend our work by including evaluation of defenses for high-resolution images, the attack by [Yin et al., 2021] (when its implementation becomes available), and more defense mechanisms including those rely on adding noise to gradients.

