## Acknowledgments

This project is supported in part by Ma Huateng Foundation, Schmidt Foundation, NSF, Simons Foundation, ONR and DARPA/SRC. Yangsibo Huang and Samyak Gupta are supported in part by the Princeton Graduate Fellowship.

We would like to thank Quanzheng Li, Xiaoxiao Li, Hongxu Yin and Aoxiao Zhong for helpful discussions, and members of Kai Li's and Sanjeev Arora's research groups for comments on early versions of the work.

## References

Martin Abadi, Andy Chu, Ian Goodfellow, H Brendan McMahan, Ilya Mironov, Kunal Talwar, and Li Zhang. Deep learning with differential privacy. In *Proceedings of the 2016 ACM SIGSAC conference on computer and communications security*, pages 308–318, 2016.

Accountability Act. Health insurance portability and accountability act of 1996. *Public law*, 104:191, 1996.

Mathieu Andreux, Jean Ogier du Terrail, Constance Beguier, and Eric W Tramel. Siloed federated learning for multi-centric histopathology datasets. In *Domain Adaptation and Representation Transfer, and Distributed and Collaborative Learning*, pages 129–139. Springer, 2020.

K. A. Bonawitz, Vladimir Ivanov, Ben Kreuter, Antonio Marcedone, H. Brendan McMahan, Sarvar Patel, Daniel Ramage, Aaron Segal, and Karn Seth. Practical secure aggregation for federated learning on user-held data. In *NIPS Workshop on Private Multi-Party Machine Learning*, 2016.

Nicholas Carlini, Samuel Deng, Sanjam Garg, Somesh Jha, Saeed Mahloujifar, Mohammad Mahmoody, Shuang Song, Abhradeep Thakurta, and Florian Tramer. An attack on instahide: Is private learning possible with instance encoding? In *IEEE Symposium on Security and Privacy*, 2020.

Sitan Chen, Xiaoxiao Li, Zhao Song, and Danyang Zhuo. On instahide, phase retrieval, and sparse matrix factorization. In *ICLR*, 2021.

Michael B Cohen, Yin Tat Lee, and Zhao Song. Solving linear programs in the current matrix multiplication time. In *Proceedings of the 51st Annual ACM Symposium on Theory of Computing (STOC)*, 2019.

European Commission. 2018 reform of eu data protection rules. https://gdpr-info.eu/, 2018.

Jia Deng, Wei Dong, Richard Socher, Li-Jia Li, Kai Li, and Li Fei-Fei. Imagenet: A large-scale hierarchical image database. In *CVPR*, 2009.

Li Deng. The mnist database of handwritten digit images for machine learning research. *IEEE Signal Processing Magazine*, 29(6):141–142, 2012.

Cynthia Dwork. The differential privacy frontier. In *Theory of Cryptography Conference (TCC)*, pages 496–502, 2009.

Cynthia Dwork and Aaron Roth. The algorithmic foundations of differential privacy. *Foundations and Trends in Theoretical Computer Science*, 9(3–4):211–407, 2014.

Jonas Geiping, Hartmut Bauermeister, Hannah Dröge, and Michael Moeller. Inverting gradients–how easy is it to break privacy in federated learning? In *NeurIPS*, 2020.

Kaiming He, Xiangyu Zhang, Shaoqing Ren, and Jian Sun. Deep residual learning for image recognition. In *CVPR*, 2016.

Yangsibo Huang, Zhao Song, Kai Li, and Sanjeev Arora. Instahide: Instance-hiding schemes for private distributed learning. In *ICML*, 2020.

Sergey Ioffe and Christian Szegedy. Batch normalization: Accelerating deep network training by reducing internal covariate shift. In *ICML*, 2015.

Peter Kairouz, H Brendan McMahan, Brendan Avent, Aurélien Bellet, Mehdi Bennis, Arjun Nitin Bhagoji, Keith Bonawitz, Zachary Charles, Graham Cormode, Rachel Cummings, et al. Advances and open problems in federated learning. *Foundations and Trends in Machine Learning*, 14(1–2): 1–210, 2021.

Diederik P Kingma and Jimmy Ba. Adam: A method for stochastic optimization. In *ICLR*, 2015.

Alex Krizhevsky et al. Learning multiple layers of features from tiny images. 2009.

Alex Lamb, Vikas Verma, Juho Kannala, and Yoshua Bengio. Interpolated adversarial training: Achieving robust neural networks without sacrificing too much accuracy. In *Proceedings of the 12th ACM Workshop on Artificial Intelligence and Security*, pages 95–103, 2019.

California State Legislature. California consumer privacy act. `https://oag.ca.gov/privacy/ccpa`, 2018.

Xiaoxiao Li, Meirui Jiang, Xiaofei Zhang, Michael Kamp, and Qi Dou. Fedbn: Federated learning on non-iid features via local batch normalization. In *ICLR*, 2021.

H Brendan McMahan, Eider Moore, Daniel Ramage, Seth Hampson, et al. Communication-efficient learning of deep networks from decentralized data. In *Artificial Intelligence and Statistics (AISTATS)*, pages 1273–1282, 2016.

Tianyu Pang, Kun Xu, and Jun Zhu. Mixup inference: Better exploiting mixup to defend adversarial attacks. In *ICLR*, 2020.

Nicolas Papernot, Steve Chien, Shuang Song, Abhradeep Thakurta, and Ulfar Erlingsson. Making the shoe fit: Architectures, initializations, and tuning for learning with privacy, 2020. URL `https://openreview.net/forum?id=rJg851rYwH`.

Le Trieu Phong, Yoshinori Aono, Takuya Hayashi, Lihua Wang, and Shiho Moriai. Privacy-preserving deep learning: Revisited and enhanced. In *ICATIS*, pages 100–110, 2017.

Le Trieu Phong, Yoshinori Aono, Takuya Hayashi, Lihua Wang, and Shiho Moriai. Privacy-preserving deep learning via additively homomorphic encryption. *IEEE Transactions on Information Forensics and Security*, 2018.

Adi Shamir. How to share a secret. *Communications of the ACM*, 22(11):612–613, 1979.

Florian Tramèr and Dan Boneh. Differentially private learning needs better features (or much more data). In *ICLR*, 2021.

Wenqi Wei, Ling Liu, Margaret Loper, Ka-Ho Chow, Mehmet Emre Gursoy, Stacey Truex, and Yanzhao Wu. A framework for evaluating gradient leakage attacks in federated learning. *arXiv preprint arXiv:2004.10397*, 2020.

Hongxu Yin, Arun Mallya, Arash Vahdat, Jose M Alvarez, Jan Kautz, and Pavlo Molchanov. See through gradients: Image batch recovery via gradinversion. *arXiv preprint arXiv:2104.07586*, 2021.

Hongyi Zhang, Moustapha Cisse, Yann N Dauphin, and David Lopez-Paz. Mixup: Beyond empirical risk minimization. In *ICLR*, 2018a.

Hongyi Zhang, Yann N Dauphin, and Tengyu Ma. Fixup initialization: Residual learning without normalization. In *ICLR*, 2019.

Richard Zhang, Phillip Isola, Alexei A Efros, Eli Shechtman, and Oliver Wang. The unreasonable effectiveness of deep features as a perceptual metric. In *CVPR*, 2018b.

Bo Zhao, Konda Reddy Mopuri, and Hakan Bilen. idlg: Improved deep leakage from gradients. *arXiv preprint arXiv:2001.02610*, 2020.

Junyi Zhu and Matthew Blaschko. R-gap: Recursive gradient attack on privacy. In *ICLR*, 2021.

Ligeng Zhu, Zhijian Liu, and Song Han. Deep leakage from gradients. In *NeurIPS*, 2019.

# A Experimental details and more results

We run all the experiments on Nvidia RTX 2080 Ti GPUs and V100 GPUs. Table 4 summarizes the set of images used in each figure or table in the main paper.

| Figure/Table | Comments |
|---|---|
| Figure 1a | We've tuned hyperparams for the attack (see Appendix A.1) and carried out evaluations on the whole CIFAR-subset. The first sampled batch of size 16 from CIFAR-subset was used in Figure 1a to demonstrate the quality of recovery for low-resolution images when BatchNorm statistics are not assumed to be known. |
| Figure 1b | We've tuned hyperparams for the attack (see Appendix A.1) and carried out evaluations on the whole ImageNet-subset. The best-reconstructed image in ImageNet-subset was used in Figure 1b to demonstrate the quality of recovery for high-resolution images when BatchNorm statistics are not assumed to be known. |
| Figure 2a | Percentages of class labels per batch were evaluated over the entire CIFAR10 dataset, for a random seed. |
| Figure 2b | The first sampled batch of size 16 was used in Figure 2b to demonstrate the quality of recovery when labels are not assumed to be known. |
| Table 2 and Figure 3 | We've tuned hyperparams for the attack and carried out evaluations on the whole CIFAR-subset. Table 2 summarizes the performance of the attack on the whole CIFAR-subset and Figure 3 shows example images. |

Table 4: Summary of experimental testbed for each evaluation.

## A.1 Hyper-parameters

**Training.** For all experiments, we train ResNet-18 for 200 epochs, with a batch size of 128. We use SGD with momentum 0.9 as the optimizer. The initial learning rate is set to 0.1 by default, except for gradient pruning with $p = 0.99$ and $p = 0.999$. where we set the initial learning rate to 0.02. We decay the learning rate by a factor of 0.1 every 50 epochs.

**The attack.** We report the performance under different $\alpha_{\mathrm{TV}}$'s (Figure 4) and $\alpha_{\mathrm{BN}}$'s (Figure 5).

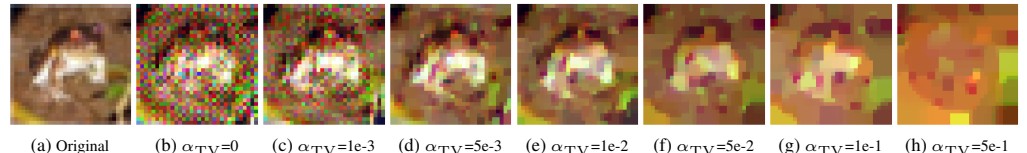

(a) Original  (b) $\alpha_{\mathrm{TV}}$=0  (c) $\alpha_{\mathrm{TV}}$=1e-3  (d) $\alpha_{\mathrm{TV}}$=5e-3  (e) $\alpha_{\mathrm{TV}}$=1e-2  (f) $\alpha_{\mathrm{TV}}$=5e-2  (g) $\alpha_{\mathrm{TV}}$=1e-1  (h) $\alpha_{\mathrm{TV}}$=5e-1

Figure 4: Attacking a single CIFAR-10 images in $\mathrm{BN}_{\mathrm{exact}}$ setting, with different coefficients for the total variation regularizer ($\alpha_{\mathrm{TV}}$'s). $\alpha_{\mathrm{TV}}$=1e-2 gives the best reconstruction.

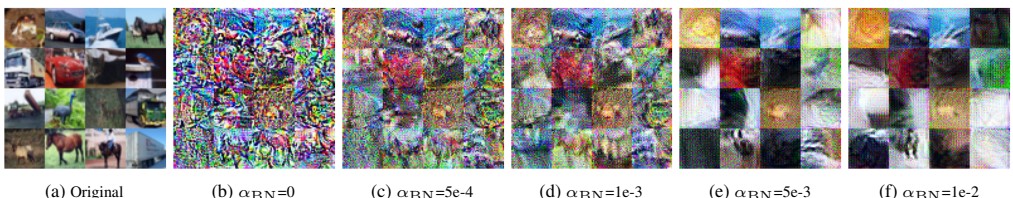

(a) Original  (b) $\alpha_{\mathrm{BN}}$=0  (c) $\alpha_{\mathrm{BN}}$=5e-4  (d) $\alpha_{\mathrm{BN}}$=1e-3  (e) $\alpha_{\mathrm{BN}}$=5e-3  (f) $\alpha_{\mathrm{BN}}$=1e-2

Figure 5: Attacking a batch of 16 CIFAR-10 images in $\mathrm{BN}_{\mathrm{infer}}$ setting, with different coefficients for the BatchNorm regularizer ($\alpha_{\mathrm{BN}}$'s). $\alpha_{\mathrm{TV}}$=1e-3 gives the best reconstruction.

## A.2 Details and more results for Section 3

**Attacking a single ImageNet image.** We launched the attack on ImageNet using the objective function in Eq. 3, where $\alpha_{\mathrm{TV}} = 0.1$, $\alpha_{\mathrm{BN}} = 0.001$. We run the attack for 24,000 iterations using Adam optimizer, with initial learning rate 0.1, and decay the learning rate by a factor of 0.1 at $3/8, 5/8, 7/8$ of training. We rerun the attack 5 times and present the best results measured by LPIPS in Figure 1.

**Qualitative and quantitative results for a more realistic attack.** We also present results of a more realistic attack in Table 5 and Figure 6, where the attacker does *not* know BatchNorm statistics but knows the private labels. We assume the private labels to be known in this evaluation, because for those batches whose distribution of labels is uniform, the restoration of labels should still be quite accurate [Yin et al., 2021]. As shown, in the evaluated setting, the attack is no longer effective when the batch size is 32 and Intra-InstaHide with $k = 4$ is applied. The accuracy loss to stop the realistic attack is only around $3\%$ (compared to around $7\%$ to stop the strongest attack) .

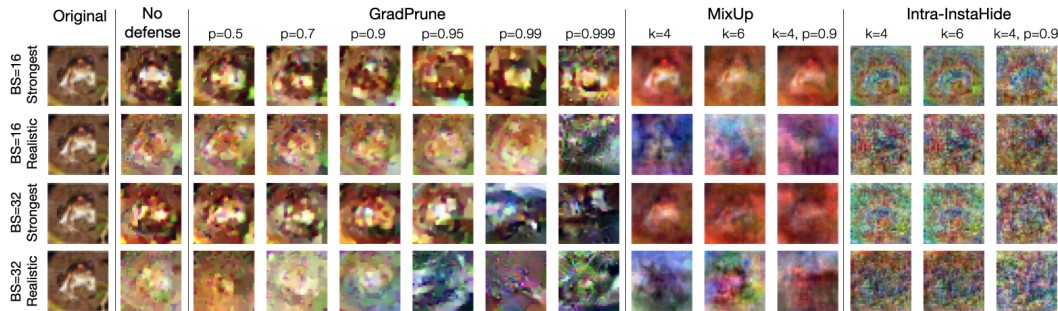

Figure 6: Reconstruction results under different defenses for a more realistic setting (when the attacker knows private labels but does not know BatchNorm statistics). We also present the results for the strongest attack from Figure 3 for comparison. Using Intra-InstaHide with $k = 4$ and batch size 32 seems to stop the realistic attack.

| | None | GradPrune ($p$) | | | | | | MixUp ($k$) | | Intra-InstaHide ($k$) | | GradPrune ($p = 0.9$) + MixUp | + Intra-InstaHide |
|---|---|---|---|---|---|---|---|---|---|---|---|---|---|
| **Parameter** | - | 0.5 | 0.7 | 0.9 | 0.95 | 0.99 | 0.999 | 4 | 6 | 4 | 6 | $k = 4$ | $k = 4$ |
| **Test Acc.** | 93.37 | 93.19 | 93.01 | 90.57 | 89.92 | 88.61 | 83.58 | 92.31 | 90.41 | 90.04 | 88.20 | 91.37 | 86.10 |
| **Time (train)** | 1× | 1.04× | | | | | | 1.06× | | 1.06× | | 1.10× | |
| | | | | | **Attack batch size** $= 16$**, the strongest attack** | | | | | | | | |
| **Avg. LPIPS** ↓ | 0.41 | 0.41 | 0.42 | 0.46 | 0.48 | 0.50 | 0.55 | 0.50 | 0.49 | 0.69 | 0.69 | 0.62 | **0.73** |
| **Best LPIPS** ↓ | 0.21 | 0.22 | 0.27 | 0.29 | 0.30 | 0.29 | 0.48 | 0.31 | 0.28 | 0.56 | 0.56 | 0.37 | **0.65** |
| (LPIPS std.) | 0.09 | 0.08 | 0.07 | 0.06 | 0.06 | 0.06 | 0.04 | 0.10 | 0.10 | 0.06 | 0.07 | 0.10 | 0.05 |
| | | | | | **Attack batch size** $= 16$**, attacker knows private labels but does not know BatchNorm statistics** | | | | | | | | |
| **Avg. LPIPS** ↓ | 0.49 | 0.51 | 0.48 | 0.51 | 0.52 | 0.56 | 0.60 | 0.71 | 0.71 | **0.75** | **0.75** | 0.74 | 0.74 |
| **Best LPIPS** ↓ | 0.30 | 0.33 | 0.31 | 0.33 | 0.34 | 0.39 | 0.44 | 0.48 | 0.53 | **0.65** | 0.63 | 0.61 | 0.63 |
| (LPIPS std.) | 0.08 | 0.09 | 0.08 | 0.08 | 0.07 | 0.07 | 0.05 | 0.08 | 0.07 | 0.04 | 0.05 | 0.08 | 0.05 |
| | | | | | **Attack batch size** $= 32$**, the strongest attack** | | | | | | | | |
| **Avg. LPIPS** ↓ | 0.45 | 0.46 | 0.48 | 0.52 | 0.54 | 0.58 | 0.63 | 0.50 | 0.49 | 0.69 | 0.69 | 0.62 | **0.73** |
| **Best LPIPS** ↓ | 0.18 | 0.18 | 0.22 | 0.31 | 0.43 | 0.48 | 0.54 | 0.31 | 0.28 | 0.56 | 0.56 | 0.37 | **0.65** |
| (LPIPS std.) | 0.11 | 0.11 | 0.09 | 0.07 | 0.05 | 0.04 | 0.04 | 0.10 | 0.10 | 0.06 | 0.07 | 0.10 | 0.05 |
| | | | | | **Attack batch size** $= 32$**, attacker knows private labels but does not know BatchNorm statistics** | | | | | | | | |
| **Avg. LPIPS** ↓ | 0.48 | 0.50 | 0.53 | 0.53 | 0.55 | 0.60 | 0.63 | 0.73 | 0.72 | 0.76 | 0.76 | 0.76 | **0.77** |
| **Best LPIPS** ↓ | 0.29 | 0.32 | 0.32 | 0.31 | 0.40 | 0.41 | 0.55 | 0.63 | 0.60 | **0.68** | 0.63 | 0.66 | 0.65 |
| (LPIPS std.) | 0.08 | 0.07 | 0.07 | 0.08 | 0.08 | 0.06 | 0.04 | 0.06 | 0.06 | 0.04 | 0.05 | 0.06 | 0.05 |

Table 5: Utility-security trade-off of different defenses for a more realistic setting (when the attacker knows private labels but does not know BatchNorm statistics). We also present the results for the strongest attack from Table 2 for comparison. We evaluate the attack on 50 CIFAR-10 images and report the LPIPS score (↓: lower values suggest more privacy leakage). We mark the least-leakage defense measured by the metric in **green**.

## A.3 More results for the strongest attack

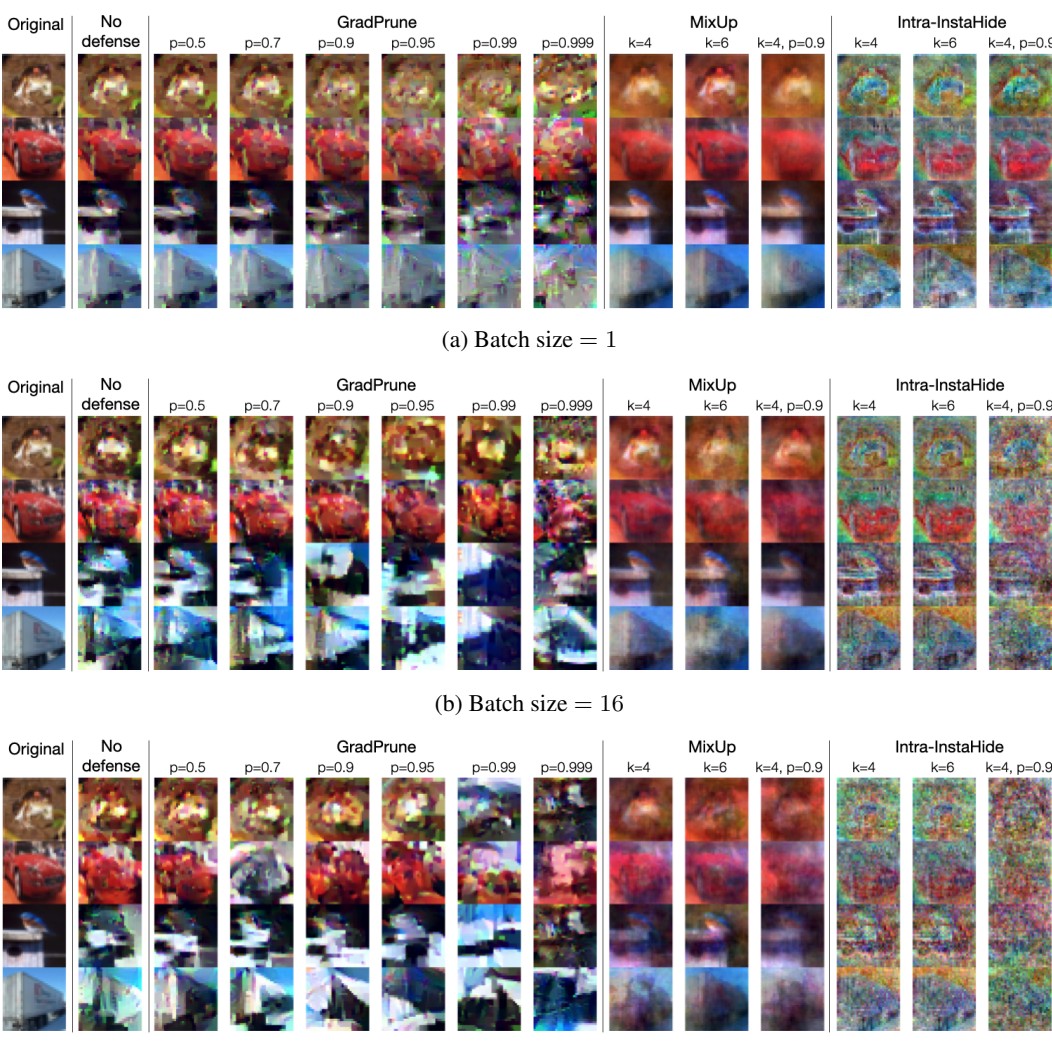

(a) Batch size = 1

(b) Batch size = 16

(c) Batch size = 32

Figure 7: Reconstruction results under different defenses with batch size 1 (a), 16 (b) and 32 (c). Full version of Figure 3.

**Results with MNIST dataset.** We've repeated our main evaluation of defenses and attacks (Table 2) on MNIST dataset [Deng, 2012] with a simple 6-layer ConvNet model. Note that the simple ConvNet does not contain BatchNorm layers. We evaluate the following defenses on the MNIST dataset with a 6-layer ConvNet architecture against the strongest attack (private labels known):

- GradPrune (gradient pruning): gradient pruning sets gradients of small magnitudes to zero. We vary the pruning ratio $p$ in {0.5, 0.7, 0.9, 0.95, 0.99, 0.999, 0.9999}.
- MixUp: we vary $k$ in {4,6}, and set the upper bound of a single coefficient to 0.65 (coefficients sum to 1).
- Intra-InstaHide: we vary $k$ in {4,6}, and set the upper bound of a single coefficient to 0.65 (coefficients sum to 1).
- A combination of GradPrune and MixUp/Intra-InstaHide.

We run the evaluation against the strongest attack and batch size 1 to estimate the upper bound of privacy leakage. Specifically, we assume the attacker knows private labels, as well as the indices of mixed images and mixing coefficients for MixUp and Intra-InstaHide.

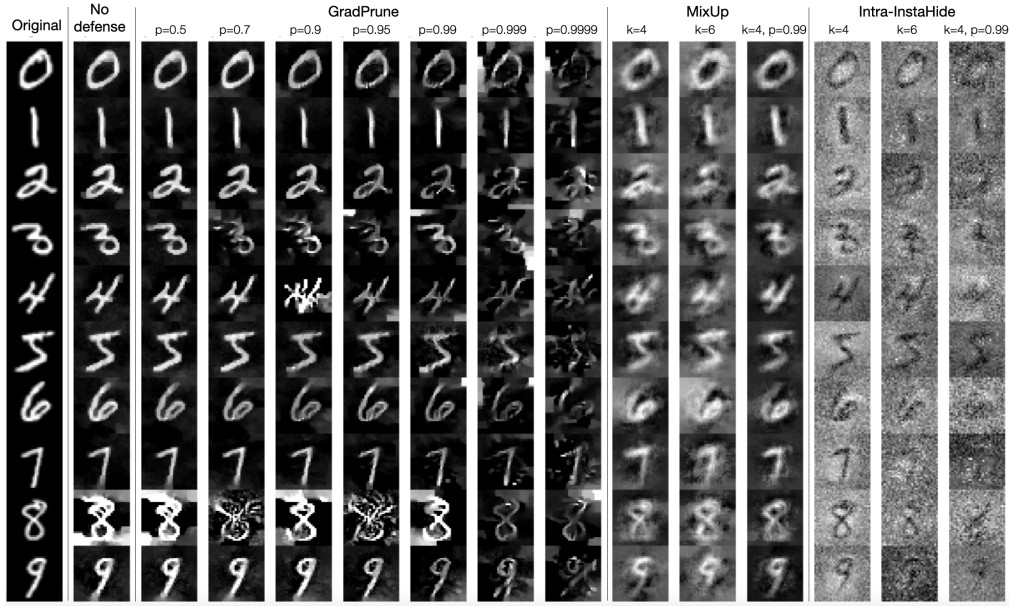

Figure 8: Reconstruction results of MNIST digits under different defenses with the strongest atttack and batch size 1.

For MNIST with a simple 6-layer ConvNet, defending the strongest attack with gradient pruning may require the pruning ratio $p \geq 0.9999$. MixUp with $k = 4$ or $k = 6$ are not sufficient to defend the gradient inversion attack. Combining MixUp ($k = 4$) with gradient pruning ($p = 0.99$) improves the defense, however, the reconstructed digits are still highly recognizable. Intra-InstaHide alone ($k = 4$ or $k = 6$) gives a bit better defending performance than MixUp and GradPrune. Combining InstaHide ($k = 4$) with gradient pruning ($p = 0.99$) further improves the defense and makes the reconstruction almost unrecognizable.

## A.4 More results for encoding-based defenses

We visualize the whole reconstructed dataset under MixUp and Intra-InstaHide defenses with different batch sizes in Figure 10, 11 and 12. Sample results of the original and the reconstructed batches are provided in Figure 9.

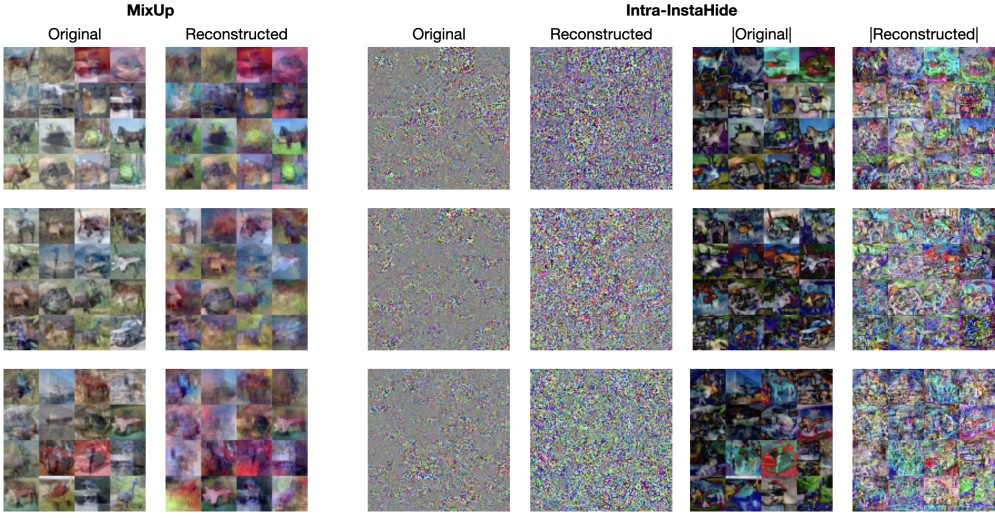

Figure 9: Original and reconstructed batches of 16 images under MixUp and Intra-InstaHide defenses. We visualize both the original and the absolute images for the Intra-InstaHide defense. Intra-InstaHide makes pixel-wise matching harder.

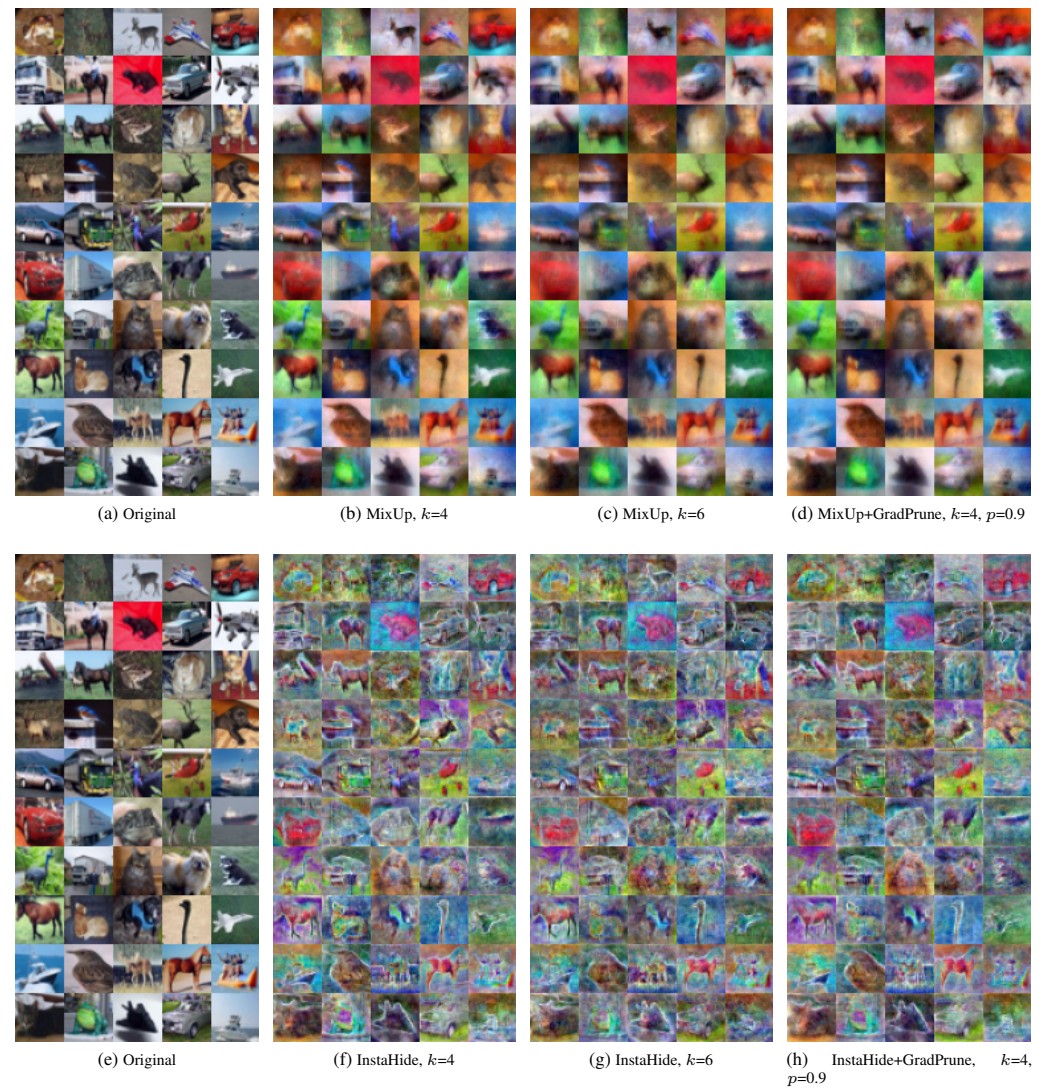

(a) Original     (b) MixUp, $k$=4     (c) MixUp, $k$=6     (d) MixUp+GradPrune, $k$=4, $p$=0.9

(e) Original     (f) InstaHide, $k$=4     (g) InstaHide, $k$=6     (h) InstaHide+GradPrune, $k$=4, $p$=0.9

Figure 10: Reconstrcuted dataset under MixUp and Intra-InstaHide against the strongest attack (batch size is 1).

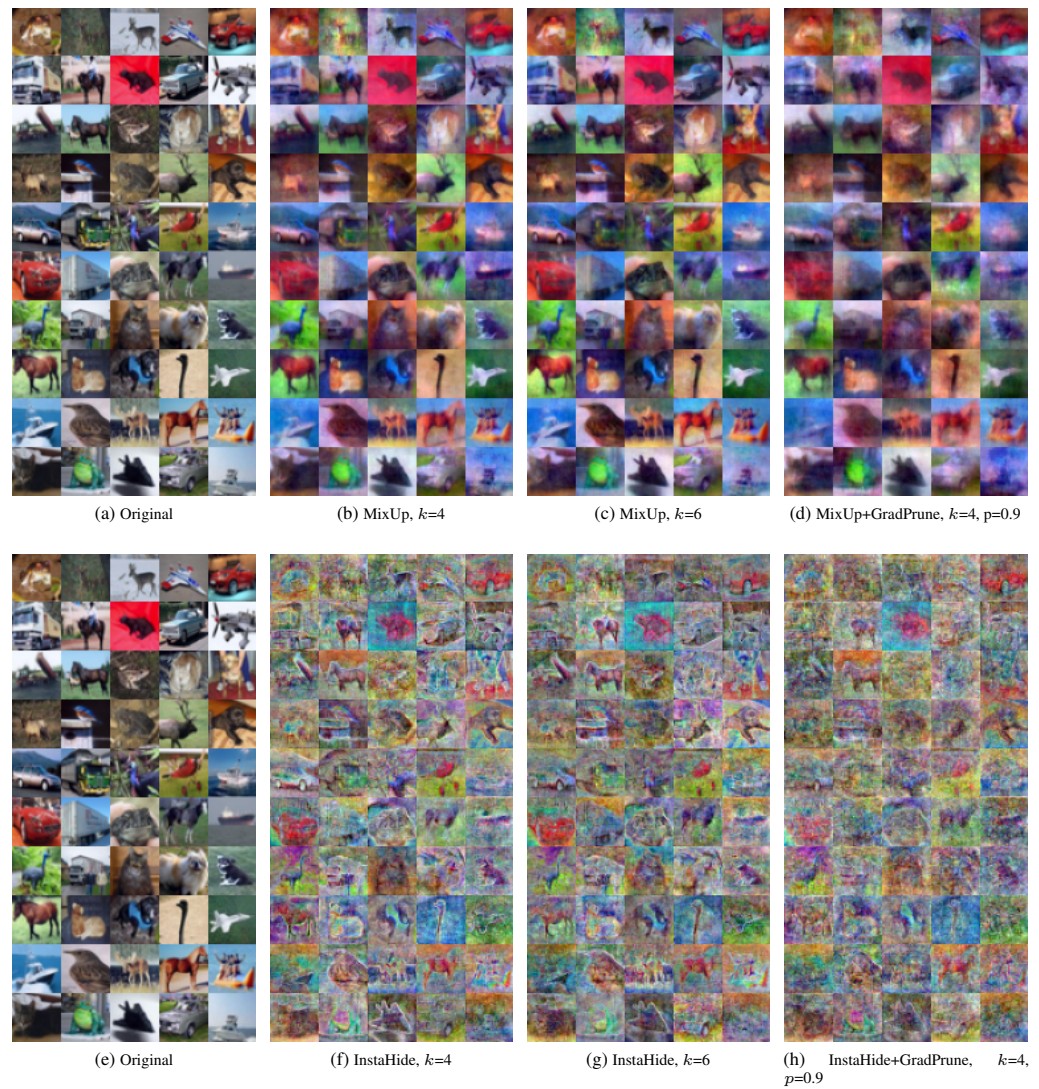

(a) Original      (b) MixUp, $k$=4      (c) MixUp, $k$=6      (d) MixUp+GradPrune, $k$=4, p=0.9

(e) Original      (f) InstaHide, $k$=4      (g) InstaHide, $k$=6      (h) InstaHide+GradPrune, $k$=4, $p$=0.9

Figure 11: Reconstrcuted dataset under MixUp and Intra-InstaHide against the strongest attack (batch size is 16).

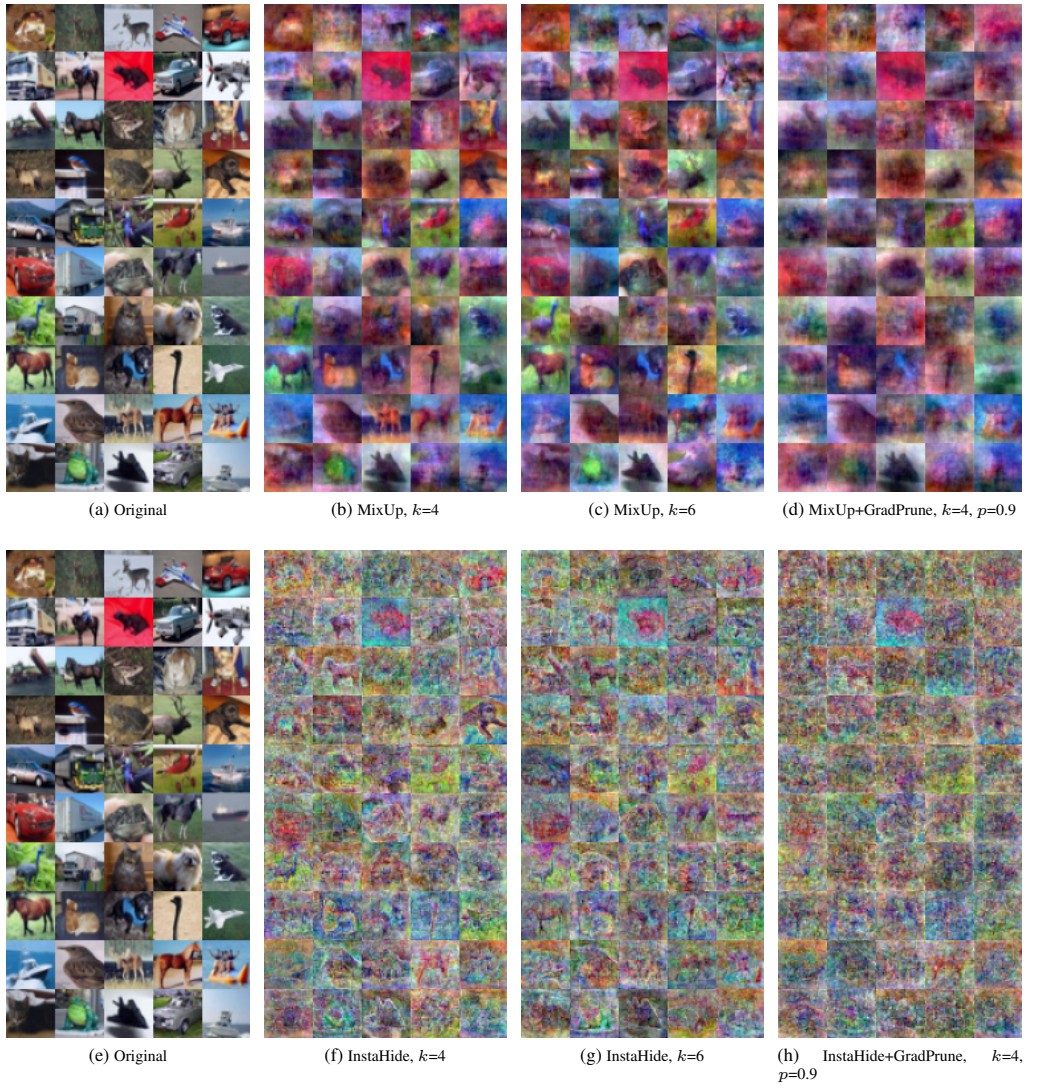

(a) Original      (b) MixUp, $k$=4      (c) MixUp, $k$=6      (d) MixUp+GradPrune, $k$=4, $p$=0.9

(e) Original      (f) InstaHide, $k$=4      (g) InstaHide, $k$=6      (h) InstaHide+GradPrune, $k$=4, $p$=0.9

Figure 12: Reconstrcuted dataset under MixUp and Intra-InstaHide against the strongest attack (batch size is 32).

# B  Theoretical Insights for Defenses' Working Mechanism

In this section, we provide some theoretical insights for the mechanism of each defense.

## B.1  Gradient pruning

Gradient pruning is a non-oblivious case of applying sketching techniques [Cohen et al., 2019] to compress the gradient vector. Usually, if we only observe the vector after sketching, it is hard to recover the original vector unless certain assumptions of the vector itself and the sketching technique have been made. Therefore, gradient pruning prevents the attacker from seeing the original gradient and make the inversion harder.

## B.2  MixUp and InstaHide

Intuitively, MixUp and InstaHide's working mechanism may come from mixing $k$ images in a single encoded image, which appears to be similar to multiplying the batch size by a factor of $k$, thus makes the attack less effective. In Section B.3, we provide theoretical analysis for this intuition, by showing that mixing $k$ images and using a batch size of $k$ are essentially similar, with any neural network that has a fully connected layer as its first layer.

Another layer of security of InstaHide seems to come from applying random sign-flipping on the mixed images. As mentioned in Section 5.1, for an InstaHide-encoded image $x \in \mathbb{R}^d$, we apply the total variation regularizer on $|x|$ instead of $x$, which pushes the gap between absolute value of adjacent pixels (i.e., $||x_j| - |x_{j+1}||$) to be small. However having $||x_j| - |x_{j+1}|| = \delta$ for some small $\delta < 10^{-4}$ does not imply that $|x_j - x_{j+1}| = \delta$; in fact, $|x_j - x_{j+1}|$ can be as large as $1 - \delta$. Therefore, the random sign flipping operation in InstaHide could potentially make the total variation image prior less effective in some sense (see Figure 9).

## B.3  Property of gradient in a small batch

The goal of this section is to present the following results,

**Lemma B.1.** *Given a neural network with ReLU activation function, each row of the gradient of first layer weights is a linear combination of images, i.e.*

$$(\frac{\partial \mathcal{L}(W)}{\partial W_1})_i = \sum_{j=1}^{b} \alpha_{i,j} x_i^{\top}$$

*where the $b$ is the number of images in a small batch, $\{x_1, \cdots, x_b\} \in \mathbb{R}^d$ are images in that small batch.*

In Section B.4 and B.5, we show the above observation holds for one/two-hidden layer neural network. In Section B.6, we generalize it to multiple layer neural network.

The standard batched $k$-vector sum can be defined as follows:

**Definition B.2.** *Give a database $X$ list of vectors $x_1, \cdots, x_n$. Given a list of observations $y_1, \cdots, y_m$ where for each $j \in [m]$, there is a set $S_j$ such that $y_j = \sum_{i \in S_j} x_i$ and $|S_j| = b$. We can observe $y_1, \cdots y_m$ but has no access to database, the goal is to recover $S_j$ and the vectors $x_i$ being use, for each $j$.*

The above definition is a mathematical abstraction of MixUp recovery/attack. It can be further generalized to InstaHide, if we only observe the $|y_j|$. We also want to remark that in the above definition, we simplify the formulation by using coefficients 1 for all vectors. It can be easily generalized to settings where random coefficients are assigned to vectors in the database for MixUp/InstaHide.

Using Lemma B.1, we notice that

**Lemma B.3.** *Under the condition of Lemma B.1, given a list of observation of gradients, and the problem recovering images is also a batched vector sum problem.*

Thus, gradient attack is essentially an variation of MixUp/Instahide attack.

## B.4 One Hidden Layer

We consider a one-hidden layer ReLU activated neural network with $m$ neurons in the hidden layer:

$$f(x) = a^\top \phi(Wx)$$

where $a \in \mathbb{R}^m$ and $W \in \mathbb{R}^{m \times d}$. We define objective function $L$ as follows:

$$L(W) = \frac{1}{2} \sum_{i=1}^n (y_i - f(W, x_i, a))^2$$

We can compute the gradient of $L$ in terms of $w_r$

$$\frac{\partial L(W)}{\partial w_r} = \sum_{i=1}^n (f(W, x_i, a) - y_i) a_r x_i \mathbf{1}_{\langle w_r, x_i \rangle}$$

Let $\widetilde{x} = \frac{1}{n} \sum_{i=1}^n x_i$,

$$\frac{\partial L(W)}{\partial w_r} = (f(W, \widetilde{x}, a) - y) a_r \widetilde{x} \mathbf{1}_{\langle w_r, \widetilde{x} \rangle}$$

Another version

$$\frac{\partial L(W)}{\partial w_r} = \sum_{i=1}^n (f(W, x_i, a) - y_i) \cdot \left( a_r \widetilde{x} \mathbf{1}_{\langle w_r, \widetilde{x} \rangle} \right)$$

## B.5 Two Hidden Layers

Suppose $a \in \mathbb{R}^m$, $V \in \mathbb{R}^{m \times d}$, $W \in \mathbb{R}^{m \times m}$. The neural network is defined as $f : \mathbb{R}^d \to \mathbb{R}$, here we slightly deviate from the standard setting and assume the input dimension is $m$, in order to capture the general setting.

$$f(x) = a^\top \phi(W \phi(Vx))$$

Consider the mean square loss

$$L(W, V, a) = \frac{1}{2} \sum_{i=1}^n |f(x_i) - y_i|^2$$

The gradient with respect to $W$ is

$$\frac{\partial L(W, V, a)}{\partial W} = \sum_{i=1}^n (f(x_i) - y_i) \underbrace{\mathrm{diag}\{\phi'(W\phi(Vx_i))\}}_{m \times m} \underbrace{a}_{m \times 1} \underbrace{\phi(Vx_i)^\top}_{1 \times m}$$

and the gradient with respect to $V$ is

$$\frac{\partial L(W, V, a)}{\partial V} = \sum_{i=1}^n (f(x_i) - y_i) \underbrace{\mathrm{diag}\{\phi'(Vx_i)\}}_{m \times m} \underbrace{W^\top}_{m \times m} \underbrace{\mathrm{diag}\{\phi'(W\phi(Vx_i))\}}_{m \times m} \underbrace{a}_{m \times 1} \underbrace{x_i^\top}_{1 \times d}$$

## B.6 The multi-layers case

The following multiple layer neural network definition is standard in literature.

Consider a $L$ layer neural network with one vector $a \in \mathbb{R}^{m_L}$ and $L$ matrices $W_L \in \mathbb{R}^{m_L \times m_{L-1}}, \cdots,$ $W_2 \in \mathbb{R}^{m_2 \times m_1}$ and $W_1 \in \mathbb{R}^{m_1 \times m_0}$. Let $m_0 = d$. In order to write gradient in an elegant way, we define some artificial variables as follows

$$
\begin{array}{llll}
g_{i,1} = W_1 x_i, & h_{i,1} = \phi(W_1 x_i), & \in \mathbb{R}^{m_1} & \forall i \in [n] \\
g_{i,\ell} = W_\ell h_{i,\ell-1}, & h_{i,\ell} = \phi(W_\ell h_{i,\ell-1}), & \in \mathbb{R}^{m_\ell} & \forall i \in [n], \forall \ell \in \{2, 3, \cdots, L\}
\end{array}
\tag{4}
$$

$$
\begin{array}{lll}
D_{i,1} = \mathrm{diag}\big(\phi'(W_1 x_i)\big), & \in \mathbb{R}^{m_1 \times m_1} & \forall i \in [n] \\
D_{i,\ell} = \mathrm{diag}\big(\phi'(W_\ell h_{i,\ell-1})\big), & \in \mathbb{R}^{m_\ell \times m_\ell} & \forall i \in [n], \forall \ell \in \{2, 3, \cdots, L\}
\end{array}
$$

where $\phi(\cdot)$ is the activation function and $\phi'(\cdot)$ is the derivative of activation function.

Let $f : \mathbb{R}^{m_0} \to \mathbb{R}$ denote neural network function:

$$f(W, x) = a^\top \phi(W_L(\phi(\cdots \phi(W_1 x))))$$

Thus using definition of $f$ and $h$, we have

$$f(W, x_i) = a^\top h_{i,L}, \quad \in \mathbb{R}, \quad \forall i \in [n]$$

Given $n$ input data points $(x_1, y_1), (x_2, y_2), \cdots (x_n, y_n) \in \mathbb{R}^d \times \mathbb{R}$. We define the objective function $\mathcal{L}$ as follows

$$\mathcal{L}(W) = \frac{1}{2} \sum_{i=1}^{n} (y_i - f(W, x_i))^2.$$

We can compute the gradient of $\mathcal{L}$ in terms of $W_\ell \in \mathbb{R}^{m_\ell \times m_{\ell-1}}$, for all $\ell \geq 2$

$$\frac{\partial \mathcal{L}(W)}{\partial W_\ell} = \sum_{i=1}^{n} (f(W, x_i) - y_i) \underbrace{D_{i,\ell}}_{m_\ell \times m_\ell} \left( \prod_{k=\ell+1}^{L} \underbrace{W_k^\top}_{m_{k-1} \times m_k} \underbrace{D_{i,k}}_{m_k \times m_k} \right) \underbrace{a}_{m_L \times 1} \underbrace{h_{i,\ell-1}^\top}_{1 \times m_{\ell-1}} \tag{5}$$

Note that the gradient for $W_1 \in \mathbb{R}^{m_1 \times m_0}$ (recall that $m_0 = d$) is slightly different and can not be written by general form. Here is the form

$$\frac{\partial \mathcal{L}(W)}{\partial W_1} = \sum_{i=1}^{n} (f(W, x_i) - y_i) \underbrace{D_{i,1}}_{m_1 \times m_1} \left( \prod_{k=2}^{L} \underbrace{W_k^\top}_{m_{k-1} \times m_k} \underbrace{D_{i,k}}_{m_k \times m_k} \right) \underbrace{a}_{m_L \times 1} \underbrace{x_i^\top}_{1 \times m_0} \tag{6}$$