# OpenReview forum: "Evaluating Gradient Inversion Attacks and Defenses in Federated Learning"
_NeurIPS.cc/2021/Conference — NeurIPS 2021 Oral_

### Official Review · Reviewer_5LCL · 2021-07-11

**Rating:** 7
**Confidence:** 3

**Summary:**

This paper evaluates existing attacks and defenses to gradient inversion attack, where eavesdropper of the protocol could steal private data from the clients. It re-evaluates attacks under relaxed assumptions, the effectiveness of the defenses, and proposes actionable ways of combining defenses under realistic assumptions to have more robust FL. The paper clarifies the state of the art in gradient inversion attacks and sheds light on best practices for good defense deployment

**Limitations And Societal Impact:**

I feel the authors should emphasize more limitations related to the number of adversarial examples generated.

**Main Review:**

# Strengths
- Understanding and systematically evaluating strengths and weaknesses in the state of the art is crucial for arising fields
- Authors adjust threat model to support the idea that gradient inversion attacks are much less realistic in practice
- The paper remains always very practical, and contains many actionable points well-summarized in the conclusions.
- Code is planned of being released

# Weaknesses
- Focused only on image classification tasks
- The experimental testbed is a bit unclear, which datasets are used in which parts
- Only 50 images on CIFAR-10 used for the re-evaluation of the defenses (line 241)

# Detailed comments

The paper studies a very important problem and takes time to re-evaluate the current state of attacks and defenses in gradient inversion attacks. Privacy is of utmost importance, and it is somehow hard to realize what are the pros/cons of existing methods. The authors conduct a very systematic and thorough evaluation, with very good and convincing arguments regarding the revision of existing attack assumptions, to help reason better about more realistic scenarios in-the-wild. Nevertheless, the paper also does a great job at finding interesting properties about more secure deployments, and greatly summarized actionable points for both practitioners and researcher.

Nevertheless, I do have a couple of major concerns that I'd like the authors to discuss a bit better.

## Experimental testbed

The datasets used in the experiments are not always fully clear. Of course authors are using CIFAR-10 (mostly) and a bit of ImageNet, but it is not always clear *how much* of it they are using for the evaluation.

For example, are Figures 1 and 2 contain some images used for the actual attacks and proof-of-concept evaluation of effectiveness in presence of relaxed assumptions. Are these the only adversarial examples generated? This sounds a bit 'small' in terms of chosen images.

On line 241, you mention:
> We use a subset of 50 CIFAR-10 images to evaluate the attack performance.

This seems quite 'limited', and possibly introducing some unintentional sampling bias in the evaluation that may threaten the empirical validity of the results.

I appreciate that the Appendix has additional settings, but I wonder how much of the study may be affected/threatened by the limited sampling of the dataset for attacks.


## Task

The paper is focused only in the **image classification task**, so it would be interesting to understand whether the same properties/conclusions would hold also in other domains where deep neural networks are successful, for example speech recognition in smart home devices.

**Time Spent Reviewing:**

5

---

> ### Author Response · Authors · 2021-08-10
> **Response to Reviewer 5LCL**
>
> We thank the reviewer for the careful review and valuable feedback of our paper.  We have tried to address all your comments in the following. Please let us know if you feel we haven’t fully addressed your comments.
>
>
> ### **Q1. Explanation for focusing on image classification task**
> > The paper is focused only in the image classification task, so it would be interesting to understand whether the same properties/conclusions would hold also in other domains where deep neural networks are successful, for example speech recognition in smart home devices.
>
> Our evaluation focused on the image classification task since most existing gradient inversion attacks [1,2,3,4] are based on this task. We agree that restricting the evaluation to a single task is a potential limitation, and have emphasized this in the conclusion section of the updated manuscript. We also believe evaluating defenses and attacks for other tasks such as speech recognition is an important future work, once mature gradient inversion attacks for these tasks are proposed.
>
>
> ### **Q2. Clarification of images used for evaluation in different parts of the paper**
> > The datasets used in the experiments are not always fully clear
>
> We thank the reviewer for suggesting us to improve the clarity of experimental testbeds in different parts of the manuscript. In our experiments, we’ve created two subsets of images for evaluation:
> - CIFAR-subset: 50 images that are randomly selected from the CIFAR-10 dataset.
> - ImageNet-subset: 50 images that are randomly selected from the ImageNet dataset.
>
> The table below summarizes the set of images used in each figure/table. We’ve clarified this issue and added this table to the appendix in our updated manuscript.
>
>
> | Figure/Table | Comments                                                                                                                                                                                                                                                                                                           |   |
> |---------------------|--------------------------------------------------------------------------------------------------------------------------------------------------------------------------------------------------------------------------------------------------------------------------------------------------------------------|---|
> | Figure 1a           | We’ve tuned hyperparams for the attack (see Appendix A) and carried out evaluations on the whole CIFAR-subset. The first sampled batch of size 16 from CIFAR-subset was used in Figure 1a to demonstrate the quality of recovery for low-resolution images when BatchNorm statistics are not assumed to be known.  |   |
> | Figure 1b           | We’ve tuned hyperparams for the attack (see Appendix A) and carried out evaluations on the whole ImageNet-subset.   The best-reconstructed image in ImageNet-subset was used in Figure 1b to demonstrate the quality of recovery for high-resolution images when BatchNorm statistics are not assumed to be known. |   |
> | Figure 2a           | Percentages of class labels per batch were evaluated over the entire CIFAR10 dataset, for a random seed.                                                                                                                                                                                                           |   |
> | Figure 2b           | We’ve tuned the hyperparams for the attack (see Appendix A). The first sampled batch of size 16 was used in Figure 2b to demonstrate the quality of recovery when labels are not assumed to be known.                                                                                                              |   |
> | Table 2 & Figure 3  | We’ve tuned hyperparams for the attack (see Appendix A) and carried out evaluations on the whole CIFAR-subset. Table 2 summarizes the performance of the attack on the whole CIFAR-subset and Figure 3 shows example images.
> |   |
>
>
>
> ### **Q3. Why we use 50 random CIFAR-10 images to evaluate the attack performance**
> > This (using a subset of 50 CIFAR-10 images to evaluate the attack performance) seems quite 'limited', and possibly introducing some unintentional sampling bias in the evaluation that may threaten the empirical validity of the results.
>
> We have evaluated the attacks and defenses on a randomly chosen subset of 50 CIFAR-10 images because running the attack on the whole dataset is computationally expensive (see analysis in Section 5.4 of our paper). The table below shows the mean and variance of the LPIPS score of the recovered images across 50-image subsets chosen by different random seeds. The closeness of the results suggests that testing on 50 CIFAR-10 images does not introduce a strong sampling bias. We will add this table to the appendix of the updated manuscript.
>
> |                     | Seed 1       | Seed 2       | Seed 3       | Seed 4       | Seed 5       |
> |---------------------|--------------|--------------|--------------|--------------|--------------|
> | LPIPS (mean +- std) | .181 +- .057 | .184 +- .065 | .189 +- .069 | .197 +- .084 | .182 +- .053 |
> | LPIPS (best)        | .090         | .065         | .069         | .086         | .081         |
> |   |
>
> > I feel the authors should emphasize more limitations related to the number of adversarial examples generated.
>
> We will mention this as a potential limitation in the conclusion section of the updated manuscript.
>
>
> *References*:
>
> [1] Ligeng Zhu, Zhijian Liu, and Song Han. Deep leakage from gradients. NeurIPS 2019.
>
> [2] Bo Zhao, Konda Reddy Mopuri, and Hakan Bilen.  iDLG:  Improved deep leakage from gradients. arXiv preprint arXiv:2001.02610, 2020.
>
> [3] Jonas Geiping, Hartmut Bauermeister, Hannah Dröge, and Michael Moeller. Inverting gradients–how easy is it to break privacy in federated learning?  NeurIPS 2020.
>
> [4] Hongxu Yin, Arun Mallya, Arash Vahdat, Jose M Alvarez, Jan Kautz, and Pavlo Molchanov.  See through gradients:  Image batch recovery via gradinversion. CVPR 2021.

---

> > ### Comment · Reviewer_5LCL · 2021-08-18
> > **Reply to rebuttal**
> >
> > Thank you for your clarifications. I confirm my positive opinion of the paper.
> >
> > It'd be great if you could then include content from this answer in the actual paper, so that these aspects are better clarified.

---

### Official Review · Reviewer_PVFu · 2021-07-16

**Rating:** 7
**Confidence:** 4

**Summary:**

The paper presents an empirical investigation of existing gradient inversion attacks and defenses in federated learning setting. The authors point out that current attacks make strong assumptions such as knowing batch norm stats or private labels, whose absence would weaken the attacks. The authors also conclude with a list of best practices of doing conducting safe federated learning via comprehensive review and experiments on current defense methods.

**Limitations And Societal Impact:**

No, the limitations and potential negative societal impact are not clear in Section 6.

**Main Review:**

This paper is well written and organized. Advancing state-of-the-art is important, but checking caveats in current methods is critical as well. Summarizing current assumptions made in current gradient inversion attacks makes it easier for future research to design fair and reasonable experiments. The practices lead by experiment results of various defenses are also valuable. However, although the conclusions are interesting, I wonder if the results are strong enough. Instead of doing experiments on a single dataset (CIFAR-10) with a single architecture (ResNet18), I would like to see more diverse settings, either easier (MNIST with MLP) or more complex (ImageNet with deeper ResNets).

**Time Spent Reviewing:**

3

---

> ### Author Response · Authors · 2021-08-10
> **Response to Reviewer PVFu**
>
> We thank the reviewer for the helpful feedback. We have tried to address all your comments in the following. Please let us know if you feel we haven’t fully addressed your comments.  We will be happy to address them further.
>
>
>
> ### **Q1. Explanation for insufficient evaluation with a more complex setting**
>
> > Instead of doing experiments on a single dataset (CIFAR-10) with a single architecture (ResNet18), I would like to see more diverse settings, either easier (MNIST with MLP) or **more complex (ImageNet with deeper ResNets)**.
>
> We thank the reviewer for pointing this out and have addressed this concern in an individual post to all reviewers: https://openreview.net/forum?id=0CDKgyYaxC8&noteId=knHerZqdADx
>
> ### **Q2. Results in an easier setting**
>
> > Instead of doing experiments on a single dataset (CIFAR-10) with a single architecture (ResNet18), I would like to see more diverse settings, either **easier (MNIST with MLP)** or more complex (ImageNet with deeper ResNets).
>
> We thank the reviewer for suggesting us to evaluate with an easier setting. We’ve repeated our main experiments of defenses and attacks (Table 1 & Figure 2 of the manuscript) on MNIST dataset with a simple 6-layer ConvNet model. We chose a simple ConvNet instead of the MLP model (which is suggested by the reviewer) because the ConvNet gives a higher test accuracy on MNIST, thus allowing us to evaluate defenses with a wider range of parameters. Note that the simple ConvNet does not contain BatchNorm layers.
>
> **2.1 Experimental setup**
> Similar to the setup in our manuscript, we evaluate the following defenses on the MNIST dataset with a 6-layer ConvNet architecture against **the strongest attack** (private labels known):
> - **GradPrune** (gradient pruning): gradient pruning sets gradients of small magnitudes to zero. We vary the pruning ratio $p$ in {$0.5, 0.7, 0.9, 0.95, 0.99, 0.999, 0.9999$}.
> - **MixUp**: MixUp encodes a private image by linearly combining it with $k-1$ other images from the training set. We vary $k$ in {$4, 6$}, and set the upper bound of a single coefficient to $0.65$ (coefficients sum to $1$).
> - **Weak-InstaHide** (W-InstaHide): we vary $k$ in {$4, 6$}, and set the upper bound of a single coefficient to $0.65$.
> A combination of GradPrune and MixUp/W-InstaHide.
>
> We run the evaluation against **the strongest attack** to estimate the upper bound of privacy leakage. Specifically, we assume the attacker knows private labels, as well as the indices of mixed images and mixing coefficients for MixUp and W-InstaHide (see lines 241~252 of our manuscript). The sample results shown below are evaluated with batch size being 1, but we are happy to include more results for larger batch sizes in the updated version of the manuscript.
>
> Same to our manuscript, we use LPIPS score as the metric, where higher values suggest more mismatch between the reconstructed image and the original image (less privacy leakage).
>
> **2.2 Results and discussion**
>
> Results of batch size 1 are shown in Table 1. We have also visualized the reconstructions, but are not allowed to upload them here due to the rebuttal guidelines. We will include both tables and figures in the updated version of our manuscript.
>
> *Table 1. Utility-security trade-off of different defenses. We train a simple ConvNet model on the whole MNIST dataset, and report the averaged test accuracy and running time of 3 independent runs. We evaluate the attack on a subset of 50 MNIST images, and report the privacy leakage under the attack measured by LPIPS (lower values suggest more leakage). We mark the least-leakage defense measured by the metric in **bold**.*
>
> |                                        	| **No defense** 	|       	|       	|       	| **GradPrune** 	|        	|         	|          	|       	| **MixUp** 	|             	|       	| **W-InstaHide** 	|             	|
> |----------------------------------------	|------------	|-------	|-------	|-------	|-----------	|--------	|---------	|----------	|-------	|-------	|-------------	|-------	|-------------	|-------------	|
> |                                        	|            	| p=0.5 	| p=0.7 	| p=0.9 	| p=0.95    	| p=0.99 	| p=0.999 	| p=0.9999 	| k=4   	| k=6   	| k=4, p=0.99 	| k=4   	| k=6         	| k=4, p=0.99 	|
> | **Test accuracy**                         	| 99.41      	| 99.35 	| 99.34 	| 99.28 	| 99.17     	| 98.63  	| 96.15   	| 89.17    	| 99.20 	| 98.80 	| 97.81       	| 99.08 	| 98.08       	| 96.88       	|
> | |
> | **Avg. LPIPS**                             	| 0.28       	| 0.28  	| 0.29  	| 0.31  	| 0.33      	| 0.37   	| 0.45    	| 0.54     	| 0.53  	| 0.55  	| 0.59        	| 0.67  	| 0.74        	| **0.76**        	|
> | **Best LPIPS**                             	| 0.10       	| 0.11  	| 0.10  	| 0.12  	| 0.19      	| 0.22   	| 0.31    	| 0.31     	| 0.40  	| 0.47  	| 0.45        	| 0.54  	| **0.64**        	| **0.64**        	|
> | **(LPIPS std)**                           	| 0.12       	| 0.12  	| 0.12  	| 0.11  	| 0.09      	| 0.08   	| 0.07    	| 0.06     	| 0.05  	| 0.04  	| 0.04        	| 0.06  	| 0.04        	| 0.05        	|
> | |
>
> Consistent with our findings in the manuscript, the new results suggest that for MNIST with a simple 6-layer ConvNet,
> - Defending the strongest attack with gradient pruning may require the pruning ratio $p ≥ 0.9999$. As a trade-off, such a high pruning ratio would introduce an accuracy loss of around 10% (see Table 1).
> - MixUp with $k=4$ or $k=6$ only have minor impacts (<1%) on test accuracy, but they are not sufficient to defend the gradient inversion attack. Combining MixUp ($k=4$) with gradient pruning ($p=0.99$) improves the defense, however, the reconstructed digits are still highly recognizable.
> - W-InstaHide alone (with $k=4$ or $k=6$) gives much better defending performance than MixUp and GradPrune (measured by the LPIPS score), but with an accuracy loss of <1.5%. Combining W-InstaHide ($k=4$) with gradient pruning ($p=0.99$) further improves the defense and makes the reconstruction almost unrecognizable.
>
>
>
> ### **Q3. Limitations and potential negative societal impact**
> > the limitations and potential negative societal impact are not clear in Section 6.
>
> We thank the reviewer for pointing this out and will discuss more limitations including limited evaluation with high-resolution images, in the updated manuscript.

---

> > ### Comment · Reviewer_PVFu · 2021-08-17
> > **Follow-up**
> >
> > Thank you for the detailed explanation and the additional experiments and analysis, these have strengthened the paper a lot. My concerns are fully addressed and I am willing to raise my score from 5 to 7.

---

### Official Review · Reviewer_XbmS · 2021-07-17

**Rating:** 6
**Confidence:** 4

**Summary:**

The authors of this paper provide a systematical analysis for the robustness of federated learning. They first show that current strongest gradient inversion attacks cannot perform well under relaxed assumptions and then demonstrate a new defense that can make the attacks less effective, even under their original assumptions. Sufficient experiments provide strong support for their conclusions.

**Ethics Review Area:**

["Responsible Research Practice (e.g., IRB, documentation, research ethics)", "I don’t know"]

**Limitations And Societal Impact:**

1. Please fix the citation (geiping2020) in Line 89 of the manuscripts.
2. I like the presentation of this paper: no fancy tricks but deliver clear and practical messages for future research. But the two parts (contribution 1 vs contribution 2, 3) are kind of separate from each other. I am suggesting the authors find a good way to combine them into a single story.
3. The experiment and analysis are mainly applied to toy datasets with low-resolution images. I am wondering whether those conclusions are still valid on high-resolution images.

**Main Review:**


1. The authors reveal that the strong assumptions in the current strongest attack may not realistic and demonstrate even the current strongest attack cannot perform well under the relaxed assumption. The conclusions encourage that future research on gradient inversion should be evaluated in a more realistic setting.
2. The authors provide some simple methods to improve the robustness of federated learning against gradient inversion attacks. They get strong results with a combination of several simple practices, which can improve the security against the strongest attack under the strong assumption.
3. The paper is well organized and easy to understand. The findings and the proposed defenses and be used to evaluate the robustness of future federated learning systems.

**Time Spent Reviewing:**

5 hours

---

> ### Author Response · Authors · 2021-08-10
> **Response to Reviewer XbmS**
>
> We thank the reviewer for the careful review and valuable feedback of our paper.  We have tried to address all your comments in the following. Please let us know if you feel we haven’t fully addressed your comments.  We will be happy to address them further.
>
> ### **Q1. Explanation for insufficient evaluation with high-resolution images**
> > The experiment and analysis are mainly applied to toy datasets with low-resolution images. I am wondering whether those conclusions are still valid on high-resolution images.
>
> We thank the reviewer for pointing this out and have addressed this concern in an individual post to all reviewers: https://openreview.net/forum?id=0CDKgyYaxC8&noteId=knHerZqdADx
>
> ### **Q2. A proposal to improve the storytelling**
> > the two parts (contribution 1 vs contribution 2, 3) are kind of separate from each other. I am suggesting the authors find a good way to combine them into a single story
>
> We agree with the reviewer that the two parts are loosely connected in the introduction. We plan to rewrite the connection of the two parts as follows:
>
> There are two important aspects in preventing gradient inversion attacks in federated learning. The first is to **use secure configurations for federated learning**, and the second is to **apply defensive mechanisms**.
>
> - In the first part of the paper, we show that the state-of-the-art gradient inversion attacks (Section 3) make two strong assumptions (sharing BatchNorm statistics and using a small batch size). By configuring or implementing a federated learning setting to ensure these assumptions no longer hold will significantly weaken the attacks (e.g. not sharing the BatchNorm statistics, and increasing the batch size or accumulate gradients of multiple steps to make it harder to infer the label).
> - In the second part of the paper, we summarize various defense methods (Section 4) and systematically evaluate (Section 5) some of their performance of defending against a state-of-the-art gradient inversion attack, and present their data utility and privacy leakage trade-offs.
>
> Finally, we will present best practices based on these two aspects.
>
> ### **Q3. Fix the typo**
>
> > Please fix the citation (geiping2020) in Line 89 of the manuscripts.
>
> We’ve fixed the typo in the updated manuscript.

---

### Official Review · Reviewer_g1LK · 2021-07-20

**Rating:** 9
**Confidence:** 4

**Summary:**

This paper has empirical study of federated learning with the following contributions:
1. It evaluates the state-of-the-art gradient attack algorithms under weaker assumptions. For example, attackers do not have access to batch norm statistics and some private labels. Under these weaker assumptions, these attack algorithms perform significantly worse.
2. It experiments various defense algorithms. For example, gradient pruning, MixUp, Weak-InstaHide, with the current state-of-the-art Carlini et al. 2020 attack algorithm. Further, it designs and evaluates a new method which combines various defense algorithms. Through empirical evaluations, it is shown that when attack batch size is 1, W-InstaHide is highly useful, and when attack batch size increases, the combined gradient pruning+W-InstaHide gives the best result.

**Limitations And Societal Impact:**

I did not realize any potential negative societal impact of the work.

**Main Review:**

Pros:
1. This paper demonstrates some practical perspective of the state-of-the-art gradient attack algorithms by removing certain strong assumptions and experimenting in more general conditions. Its empirical result is strong and clear that as soon as these strong assumptions are removed, the SOTA attack algorithms no longer perform that well. That sheds lights on how to improve the security of a federated learning system, i.e., do not share statistics like batch norm or certain private labels.
2. From a defense perspective, this paper considers a novel scheme which combines various different defense algorithms, such as gradient pruning, MixUp and InstaHide to achieve a better defense performance compared to any single defense algorithm, especially when attack batch size is large.
3. Previous works either focus on encoding private dataset directly or encoding the gradient information. This paper is a unified review of these two different methods, and combines them to improve the overall defense performance.

Cons:
The line 233 is a bit confusing: “InstaHide further encodes a MixUp image by randomly flipping its pixel signs. We are only mixing private images in our experiments, which is a weaker version of InstaHide”. Does this mean that W-InstaHide does not use random sign flips? If so, what’s the difference between MixUp and W-InstaHide? I believe there should be a difference between MixUp and W-InstaHide. However, based on the write-up, it’s a bit unclear about such differences and hence confusing. It should be further clarified.

**Time Spent Reviewing:**

6 hours

---

> ### Author Response · Authors · 2021-08-10
> **Response to Reviewer g1LK**
>
> We thank the reviewer for your careful review and detailed summary of our manuscript. We have tried to address all your comments in the following. We highly appreciate knowing if our responses have addressed your initial comments.
>
> > The line 233 is a bit confusing ... based on the write-up, it’s a bit unclear about such differences and hence confusing. It should be further clarified.
>
> We will clarify this. InstaHide paper (Huang et al. 2020) proposes two versions: Inter-InstaHide and Intra-InstaHide.  The only difference is that at the mixup step, Inter-Instahide mixes up an image with images from a public dataset, whereas Intra-InstaHide mixes with private images.   Both versions apply a random sign flipping pattern on each mixed image. The latter is called W-InstaHide in our submission. To reduce confusion and be consistent with (Huang et al. 2020), we have rewritten the sentence, and changed W-InstaHide to Intra-InstaHide in our updated version.

---

### Author Response · Authors · 2021-08-10
**Thanks for your valuable feedback and a summary of comments**

We thank AC and all reviewers for their time and valuable feedback, and for the recognition that our manuscript "*demonstrates some practical perspective of the state-of-the-art gradient attack algorithms*", and the "*empirical result is strong and clear*". Reviewers also highlight that our evaluation "*contains many actionable points*", and ​"*delivers clear and practical messages for future research*".



### **A common concern (Reviewer XbmS Q2 & Reviewer PVFu Q1): evaluation is mainly on CIFAR-10; how about  high-resolution images?**


We've used both CIFAR-10 and ImageNet in the evaluation of strong assumptions (Section 3). However, the current SOTA attacks mainly work on low-resolution images and achieve poor performance on high-resolution images (see Figure 1). Therefore evaluation on low-resolution images makes the most sense in trying to understand whether current attacks can be mitigated.

The updated manuscript now notes that a systematic evaluation of high-resolution images will be important future work **once stronger attacks are proposed.** We are also happy to incorporate new attacks into our current implementation and evaluations/comparisons.

### **Individual comments**
We have summarized other comments and our responses as below (corresponding reviewer IDs are provided in parenthesis). Detailed responses can be found in posts to individual reviewers.

1. **Clarification about the experimental setup**: we’ve clarified the experimental  testbed for each evaluation in the manuscript (Reviewer 5LCL, Q2). We also showed that the selected testbeds do not introduce a strong sampling bias (Reviewer 5LCL, Q3).

2. **Presenting results in an easier setting**: we’ve evaluated attacks and defenses with an easier setting: MNIST + a simple ConvNet (Reviewer PVFu, Q2). The results are consistent with our findings in the manuscript.

3. **Improving the storytelling**: we’ve proposed a way to improve the current writing to better connect the two parts (contribution 1 vs contribution 2, 3) of the manuscript (Reviewer XbmS, Q2).

4. **Discussing more future work**: we’ve discussed more future works in the updated manuscript, including extending the evaluation to high-resolution images (Reviewer XbmS Q2 & Reviewer PVFu Q1), or to language tasks (Reviewer 5LCL, Q1).

5. We’ve also fixed some typos (Reviewer XbmS, Q3) and rewritten sentences that may cause confusion (Reviewer g1LK, Q1).

---

### Decision · Program_Chairs · 2021-09-27

**Decision:**

Accept (Oral)

**Comment:**

The reviewers are satisfied by the responses made by the authors. The authors are strongly encouraged to include the additional experiments and results they provided in the rebuttal phase to their final manuscript.